# Higher-Order Causal Structure Learning with Additive Models

## Abstract

Causal structure learning has long been the central task of inferring causal insights from data. Despite the abundance of real-world processes exhibiting higher-order mechanisms, however, an explicit treatment of interactions in causal discovery has received little attention. In this work, we focus on extending the causal additive model (CAM) to additive models with higher-order interaction. This second level of modularity we introduce to the structure learning problem is most easily represented by a directed acyclic hypergraph. We introduce the necessary definitions and theoretical tools to handle the novel structure we introduce and then provide identifiability results for the hyper DAG, extending the typical Markov equivalence classes. We next provide insights into why learning the more complex hypergraph structure may actually lead to better empirical results. In particular, more restrictive assumptions like CAM correspond to easier-to-learn hyper DAGs and better finite sample complexity. We finally develop an extension of the greedy CAM algorithm which can handle the more complex hyper DAG search space and demonstrate its empirical usefulness in synthetic experiments.

## 1 Introduction

Causal structure learning aims to infer the underlying causal relationships among given variables from observational or interventional data [Spirtes et al., 2001], which is crucial for understanding complex systems and has been widely applied in different fields, including biology [Sachs et al., 2005] and Earth system science [Runge et al., 2019]. Various approaches have been developed for causal discovery, including constraint-based, score-based, and functional causal model-based methods [Glymour et al., 2019].

Constraint-based methods, such as the PC [Spirtes and Glymour, 1991] and FCI [Spirtes et al., 2001] algorithms, rely on conditional independence tests to identify the causal structure. Score-based methods, on the other hand, optimize a scoring function, such as the Bayesian Information Criterion (BIC) [Schwarz, 1978], to find the best causal structure [Chickering, 2002, Singh and Moore, 2005, Yuan et al., 2011, Bartlett and Cussens, 2017]. Both constraint-based and score-based approaches can only identify the underlying causal structure up to Markov equivalence [Spirtes et al., 2001], indicating that they cannot distinguish between different structures that encode the same set of conditional independence relationships.

Functional causal model-based methods address this limitation by introducing proper functional assumptions on the causal relationships, thus enabling the identification of the whole DAG. Examples include the linear non-Gaussian model [Shimizu et al., 2006], additive noise model (ANM) [Hoyer et al., 2008], post-nonlinear causal model [Zhang and Hyvärinen, 2009], heteroscedastic noise model (HNM) [Xu et al., 2022, Immer et al., 2023], and causal additive model (CAM) [Bühlmann et al., 2014]. Among these, CAM assumes that the causal relationships are additive in the variables, which,

despite being more restrictive than the general ANM framework, has been shown to achieve superior performance in various empirical studies [Lachapelle et al., 2020, Zheng et al., 2020, Ng et al., 2022, Rolland et al., 2022], partly owing to its improved statistical power.

In this work, we revisit the additive structural assumption of CAM by incorporating recent developments in training higher-order additive models, extending the functional causal model to explicitly consider the higher-order interactions within the causal mechanisms. Higher-order mechanisms are known to exist in a variety of real-world processes and are believed to be critical into modeling and understanding a number of different scientific phenomena Battiston et al. [2020], Majhi et al. [2022]. Nevertheless, previous approaches have taken an all-or-nothing approach, either (a) directly following CAM-like assumptions or (b) modeling all possible interactions between parents of a child node.

Instead, we find that a directed hypergraph can succinctly represent the necessary structure to interpolate between the simplicity of CAM and the complexity of the full ANM. Specifically, our major contributions are as follows:

1. We develop the theoretical extension from graphs to hypergraphs across three total settings (undirected graphical models, classical DAG models, and additive noise models), and prove the identifiability of the hypergraph structures we introduce.

2. We develop an algorithm for learning the hyper DAG alongside its structural equations directly from data, extending the greedy algorithm for CAM, and showing improved performance over existing approaches on data specifically containing higher-order variable interactions.

## 2 Hypergraph Methods

In this work we will introduce three different generalizations to existing structure learning approaches which extends the existing graphical representations (Markov networks, Bayesian networks, etc.) to their corresponding hypergraphical representations:

1. Undirected hypergraphical models

2. Directed hypergraphical models for discrete variables (classical regime)

3. Directed hypergraphical models for continuous variables (additive noise model)

We will first introduce the 'hyper Markov property' which will be respected by distributions which are 'Markov' with respect to a given hypergraph, rather than Markov with respect to a given graph. We emphasize that since hypergraphs are a strict generalization of existing graphical models, we can see this hyper DAG or HDAG structure as an intermediate level of structure between the DAG and the SEM (structural equation model). In that sense, we write:

$$\text{DAGs} \preccurlyeq \text{HDAGs} \preccurlyeq \text{SEMs} \tag{1}$$

In what follows, we will demonstrate that this more fine-grained structure is not only identifiable directly from data, but also that this perspective allows for greater insights into the identifiability of different hypergraphs (and hence graphs) using finite observations rather than the population limit.

### 2.1 Undirected Models

Let us write $X \in \mathbb{R}^d$ for some number of dimensions $d \in \mathbb{N}$. We will later choose to restrict to discrete, continuous, or mixed $X$ as appropriate. We write an *undirected* graph as $\mathcal{G}' = (V, E')$ and *undirected* hypergraph as $\mathcal{H}' = (V, H')$, where we take the vertices as $V = [d] := \{1, \ldots, d\}$, the undirected edges as $E' \subseteq \{(i, j) : i \neq j \in V\}$, and the undirected hyperedges as $H' \subseteq \{S \subseteq V\}$. We will sometimes abuse notation and write $(i, j) \in \mathcal{G}'$ to mean $(i, j) \in E'$ and similarly for $\mathcal{H}'$. (Note that we are reserving the unprimed versions for the directed versions.)

We will assume throughout this work that we are in the case of fully observed variables. Moreover, we will assume that the density is strictly positive to ensure (a) that there is no confusion caused by switching between the pairwise, local, and global properties; and (b) that the score-based definitions we introduce on the log-probability face no ambiguities in regions of zero density.

**Definition 1.** *Undirected Markov Property*. Let us take $N(i)$ to denote the neighbors of $i \in V$. We may say that some distribution $p_X(\boldsymbol{x})$ is (locally) Markov with respect to some undirected graph $\mathcal{G}'$ if it holds for any $i$ that "$X_i \perp\!\!\!\perp X_{V-N(i)-\{i\}} \mid X_{N(i)}$", where – denotes set minus. Preparing for our focus on additive models of the log probability, this can equally be required as:

$$p_X(\boldsymbol{x}) = p_{N(i)}(\boldsymbol{x}_{N(i)}) \cdot p_i(x_i|\boldsymbol{x}_{N(i)}) \cdot p_{V-N(i)-\{i\}}(\boldsymbol{x}_{V-N(i)-\{i\}}|\boldsymbol{x}_{N(i)}) \quad (2)$$

$$\xi_X(\boldsymbol{x}) := \log p_X(\boldsymbol{x}) = \xi_{N(i)}(\boldsymbol{x}_{N(i)}) + \xi_i(x_i|\boldsymbol{x}_{N(i)}) + \xi_{V-N(i)-\{i\}}(\boldsymbol{x}_{V-N(i)-\{i\}}|\boldsymbol{x}_{N(i)}) \quad (3)$$

where there exists some conditional probabilities $p_i$ and $p_{V-N(i)-\{i\}}$ or some conditional log probabilities $\xi_i$ and $\xi_{V-N(i)-\{i\}}$ such that these equations hold true. This can be additionally written in terms of the clique representation, when we write all cliques of the graph as $Cl(\mathcal{G}') = \{S \subseteq V : S \text{ is a clique in } \mathcal{G}'\}$, as follows:

$$\log p_X(\boldsymbol{x}) =: \xi_X(\boldsymbol{x}) = \sum_{S \in Cl(\mathcal{G}')} \xi_S(x_S) \quad (4)$$

**Definition 2.** *Undirected Hyper-Markov Property*. It is now straightforward to generalize this property to hypergraphs as follows:

$$\log p_X(\boldsymbol{x}) =: \xi_X(\boldsymbol{x}) = \sum_{S \in \mathcal{H}'} \xi_S(x_S) \quad (5)$$

That is, we write the hypergraph edges as specifically representing the energy terms in the log-probability function. It is straightforward to verify that this is strictly more general than hypergraphs which can be created as a result of the maximal clique structure of a typical graph. Nonetheless, in what follows we hope to focus on the identifiability as well as the usefulness of this finer-grained structure for graphical models.

## 2.2 Directed Classical Models

We will write a directed graph as $\mathcal{G} = (V, E)$ and a directed hypergraph as $\mathcal{H} = (V, H)$ where the directed edges are $E \subseteq \{(k, j) : k \neq j \in V\}$ and the directed hyperedges are $H \subseteq \{(S, j) : j \in V, S \subseteq (V - j)\}$. That is, we are assuming that each hyperedge has only one "out arrow" and up to $|S|$ "in arrows". It is hoped the purpose for this is relatively clear in the context of a causal diagram which must use several parents to generate a single child. We write the 'parents of $j$ in $\mathcal{G}$' as $\text{Pa}_{\mathcal{G}}(j) = \{k \in [d] : (k, j) \in \mathcal{G}\}$ and the 'hyperparents of $j$ in $\mathcal{H}$' as $\text{HypPa}_{\mathcal{H}}(j) = \{S : (S, j) \in \mathcal{H}\}$, where the dependence on $\mathcal{G}$ and $\mathcal{H}$ will be dropped when obvious.

**Definition 3.** *Directed Markov Property*. Here, we may once again recall the classical Markov property with respect to a DAG to be written as:

$$\log p(\boldsymbol{x}) = \sum_{i=1}^{d} \log p(x_i|\boldsymbol{x}_{Pa(i)}) = \sum_{i=1}^{d} \theta(x_i|\boldsymbol{x}_{Pa(i)}) \quad (6)$$

It is very easy to see that we may rewrite this using extraneous functions as:

$$\log p(\boldsymbol{x}) = \sum_{i=1}^{d} \sum_{S \subseteq \text{Pa}(i)} \theta(x_i; \boldsymbol{x}_S) - \mathcal{Z}(\boldsymbol{x}_{\text{Pa}(i)}) \quad (7)$$

where it is now the case that we do not have the $\theta$ energy terms explicitly representing a conditional distribution, but are instead arbitrary functions which are then set to the proper normalization via the $\mathcal{Z}$ function. It can be seen that the $\mathcal{Z}$ function does not explicitly depend on the value of $x_i$, but normalizes to a distribution based on only the parents alone. The extraneous $\theta$ functions which are written as all subsets are useful for the next step generalizing to hypergraph structures.

**Definition 4.** *Directed Hyper-Markov Property*. We follow the structure above from the typical DAG framework, but replace the fully-connected parent structure with the more nuanced hypergraph structure. In particular, the energy terms in each of the conditional distributions are replaced with a more specific additive model structure, rather than assuming there is a generic function:

$$\log p(\boldsymbol{x}) = \sum_{i=1}^{d} \sum_{S \in \text{HypPa}(i)} \theta(x_i; \boldsymbol{x}_S) - \mathcal{Z}(\boldsymbol{x}_{\text{Pa}(i)}) \quad (8)$$

It is again straightforward to see that this strictly generalizes the cases which are representable by the typical DAG. In particular, taking the hyperparents to be all subsets of the parents recovers the previous functional form (Figure 1f). However, other structures mimicking CAM and LiNGAM type assumptions are also possible (Figure 1d). Further HDAGs beyond these two existing in the literature are also possible (Figure 1e). We will further assume 'causal minimality' of the HDAG meaning the hyperparent set is downwards closed w.r.t subsets and each maximal element has a nontrivial $\theta$ function. See the discussion and proofs in the appendix for further details.

It is also relatively clear to see how this *directed* hyper-Markov property overlaps with the *undirected* hyper-Markov property, perhaps moreso than the typical Markov properties. Moreover, it becomes clear that the moralized graph corresponds to including the $\mathcal{Z}$ terms whereas the skeleton corresponds to including only the $\theta$ terms, see also Table 1. We will make this connection more clear in Section 3.2, where we show identifiability of the HDAG up to its hyper Markov equivalence class (HMEC).

## 2.3 Directed Additive Noise Models

For continuous variables, we will generate data from the additive noise model (ANM), meaning that all variables are a deterministic function of their parent variables, plus an additive noise term.

**Definition 5.** *Additive Noise Model*. This may be written as:

$$X_j = f_{\text{Pa}(j) \to j}(X_{\text{Pa}(j)}) + \varepsilon_j \tag{9}$$

Each of the $\varepsilon_j$ are taken to be independent, mean-zero random variables.

**Definition 6.** *Higher-Order Additive Noise Model*. We may once again generalize to the higher-order additive model through the use of the structure encoded by the directed hypergraph.

$$X_j = \Big( \sum_{S \in \text{HypPa}(j)} f_{S \to j}(x_S) \Big) + \varepsilon_j \tag{10}$$

Specifically, we endow the generating function $f_{\text{Pa}(j) \to j}$ with an additive model structure obeying the hyperparents of the HDAG. Models like CAM or LinGAM then correspond to using singleton hyperparents, whereas the most general ANM corresponds to using the entire block of parents as the largest hyperparent, as depicted in Figure 1. We will follow CAM Bühlmann et al. [2014] in assuming Gaussian noise for algorithmic purposes via the minimization of mean-squared error corresponding to maximizing the log-likelihood; however, surprisingly, we show in Theorem 4 that the our settings 2 and 3 actually overlap in the case of additive Gaussian noise.

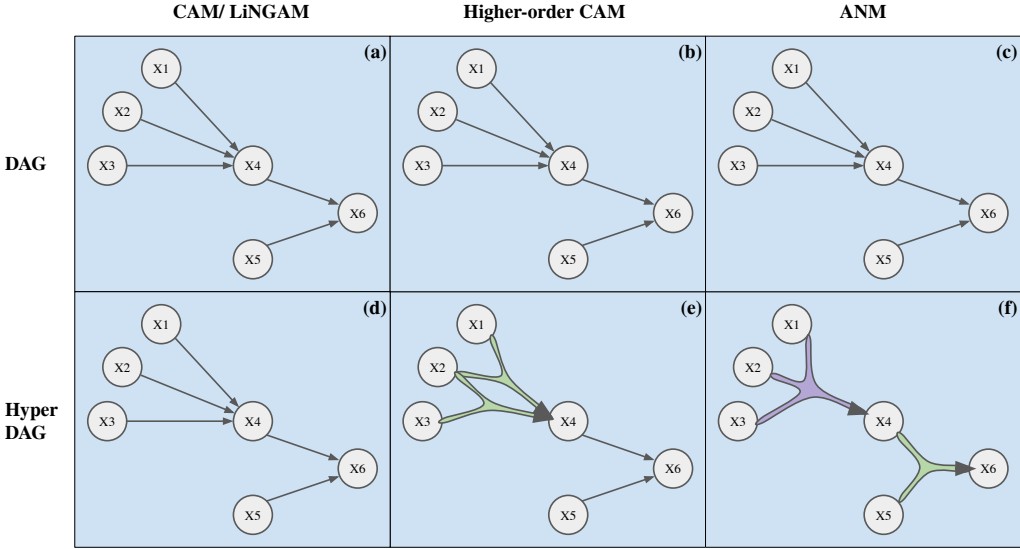

Figure 1: A depiction of the distinguishing power for hypergraphs corresponding to the same DAG.

## 3 Structure Identifiability

### 3.1 Undirected Models

First recall that under our assumptions of fully observed variables and strictly positive density, meaning that the density function is identifiable directly from the observed distribution (under mild assumptions like continuity for the continuous variable case) Rosenblatt [1956].

Importantly, then, one may only be concerned in measuring the hypergraph structure as described in Section 2.1; however, this proves to be equally straightforward. For the case of graphical models and mixed-type variables, Zheng et al. [2023] write the generalized precision matrix as:

$$\Omega_{i,j} := \left\| \frac{\partial^2}{\partial_i \partial_j} \log p(x) \right\| := \left( \mathbb{E}\left[ \left| \frac{\partial^2 \log p(x)}{\partial_i \partial_j} \right|^2 \right] \right)^{\frac{1}{2}} \tag{11}$$

In the case of hypergraphical models and discrete variables, Enouen and Sugiyama [2024] similarly write the existence of 'higher-order information' for some subset $T \subseteq [d]$ (where $T \supsetneq \{i,j\}$ is chosen to imply higher-order) if it is the case that:

$$\Omega_T := \left\| \sum_{S \supseteq T} \theta_S(x_S) \right\| > 0 \qquad \text{where} \qquad \log p(x) = \sum_S \theta_S(x_S) \tag{12}$$

A straightforward combination of these approaches are sufficient for recovery of the hyper Markov network or undirected hypergraph. Nonetheless, our experiments will instead focus on identification of the directed structure as in the following two sections. Thus, for our purposes it is sufficient to say that the density and log density functions are identifiable directly from the observed distribution.

### 3.2 Directed Classical Models

Before our main theorem of identifiability extending the result of Verma and Pearl [1990], we must first introduce the notion of multi-dependence to extend the typical notion of conditional independence which is the workhorse of causal structure learning. We will focus on discrete and finite variables as in the classical case[Verma, 1993, Pearl, 2009]; however, most results clearly extend to continuous or mixed variables under mild conditions, and we later discuss one such special case in Theorem 4.

**Definition 7.** *Conditional Multi-dependence.* We write that $X_i$ and $X_j$ are dependent if the distribution $\log p(x_i, x_j)$ must be written with a 2D energy term, $\theta_{ij}(x_i, x_j)$, rather than the sum of two 1D energy terms, $\theta_i(x_i) + \theta_j(x_j)$, (corresponding to the product when the log is removed). We will write that $X_i$, $X_j$, and $X_k$ are tri-dependent (or generally multidependent), if the distribution $\log p(x_i, x_j, x_k)$ must be written with a 3D energy term, rather than the sum of three 2D energy terms. It can be seen that this does not have a convenient product formulation like the classical case of dependence and independence because of the "mixing" or "torsion" between the three 2D terms. Nonetheless, we will attempt to prove the usefulness of such a definition in the following theorem. Generalization to conditional tests is straightforward.

**Theorem 1.** The HDAG is identifiable up to the hyper Markov Equivalence classes (HMECs), consisting of all HDAGs with the same "body" and the same (unshielded) "multi-colliders", paralleling the existing result identifying DAGs up to their skeleton and (unshielded) colliders.

In the same sense that a conditional independence test can never eliminate a causal arrow between two variables, a conditional multi-independence test can never separate a higher-order causal relationship between a set of three or more variables. Removing the arrowheads from the DAG returns the

Table 1: Notation for hypergraphs

| DAG terms | HDAG terms |
|---|---|
| $\mathcal{G}'$, undirected graph | $\mathcal{H}'$, undirected hypergraph |
| $\mathcal{G}$, directed acyclic graph (DAG) | $\mathcal{H}$, hyper DAG or HDAG |
| moralized graph of a DAG | immoralized hypergraph of an HDAG |
| skeleton of a DAG | body of an HDAG |
| (unshielded) collider | (unshielded) multicollider |

DAG's skeleton; similarly, removing the arrowheads from the HDAG returns the HDAG's body, see Table 1 and Figure 2. In some sense, this half of the theorem about the "body identifiability" immediately states that the structure we introduced is identifiable.

For multicolliders, recall that a collider occurs when there is a conditional dependence which is broken after marginalizing out the child, or equally a conditional independence which is broken when conditioning on the child. The multicollider of an HDAG will occur similarly via a multidependence which is broken after marginalizing out the child. Although collisions between two parents will already be covered, there are cases of three or more parents which are unshielded and can hence be identified from Theorem 1. In particular, there are cases which are not identified in the classical setting, see the RHS of Figure 2. This seeming anomaly is in part due to the historical conflation over time between what structure is recoverable from the conditional independence tests vs. what structure is recoverable from the observed distribution. Indeed, the MEC only describes what is distinguishable via the conditional independence conditions, making it unable to detect what can be seen via the conditional multi-independence test we introduce.

Another key consequence of this different perspective will be a statistical one. In particular, for a $K$-dimensional energy term in the body of an HDAG, we know that it requires on the order of $\mathcal{O}(n^K)$ samples to be appropriately learned. Consequently, without access to infinite samples, this places further restrictions on the HMEC classes (and hence MEC classes) of 'distinguishability under finite samples', whereas MECs are only able to easily represent 'distinguishability under infinite samples' as in the asymptotic regime.

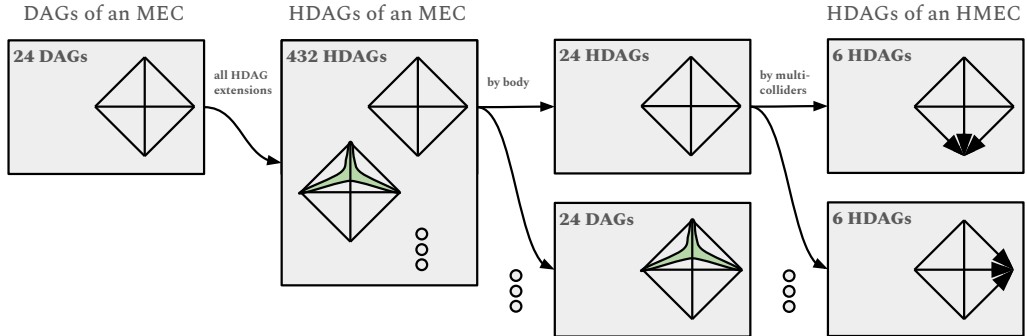

Figure 2: A gradual refinement of the DAGs within a Markov Equivalence Class (MEC) to a stronger refinement of HDAGs based on Theorem 1 to a Hyper Markov Equivalence Class (HMEC). There are $d = 4$ variables with a fully-connected DAG structure. The green triangle represents the third-degree hyperedge in the body of an HDAG. Lack of arrows indicate multiple possible orientations for different DAGs/ HDAGs of the same MEC/ HMEC.

## 3.3 Directed Additive Noise Models

In this section, we establish identifiability results for recovering the hyper-DAG in the ANM case. For clearer exposition, we first reproduce the arguments of Hoyer et al. [2008] which shows that, in general position, the additive noise model (ANM) is identifiable. We extend their result to a multi-dimensional result which handles the case of multiple parents rather than only the case of one parent node and one child node (slightly different from the extension in Theorem 28 of Peters et al. [2014] because it will more easily generalize to the hypergraph result).

**Theorem 2.** Let the joint probability densities of $\boldsymbol{x}$ and $y$ be given by

$$p(x_1, \ldots, x_d, y) = p_n(y - f(\boldsymbol{x})) \cdot p_x(x_1, \ldots, x_d) \tag{13}$$

for some noise density $p_n$, arbitrary density $p_x$, and function $f$. Assume further that all such functions are thrice continuously differentiable. If there is also a backwards model which treats $x_i$ as the child,

$$p(x_1, \ldots, x_d, y) = p_{\tilde{n}}(x_i - g(x_{-i}, y)) \cdot p_{x_{-i}, y}(x_1, \ldots, x_{i-1}, x_{i+1}, \ldots, x_d, y) \tag{14}$$

for some alternate noise density $p_{\tilde{n}}$, arbitrary density $p_{x_{-i}, y}$, and function $g$, then it must be case that for $\nu := \log p_n$ and $\xi := \log p_x$, the following differential equation is obeyed

$$\xi''' = \xi'' \cdot \Big( -\frac{\nu''' f'}{\nu''} + \frac{f''}{f'} \Big) + \Big( -2\nu'' f'' f' + \nu' f''' + \frac{\nu' \nu''' f' f''}{\nu''} - \frac{\nu' f'' f''}{f'} \Big) \tag{15}$$

where we use $\xi', \nu', f'$ as shorthand for $\frac{\partial}{\partial x_i} \xi(x_1, \ldots, x_d)$, $\nu'(y - f(\boldsymbol{x}))$, $\frac{\partial}{\partial x_i} f(x_1, \ldots, x_d)$.

**Corollary 1.** Assume further that $\nu''' = 0$ and $\frac{\partial^3}{\partial x_i^3}\xi = 0$ for all $i \in [d]$. If a backwards model exists for a parent $x_i$, then $f$ is linear in the argument $x_i$. Further, if a backwards model exists in all parents $x_i$, then $f$ is a multilinear function.

**Remark 1.** First, this is a general position argument which says that in order to be reversible, the SEM must obey these particular constraints which usually do not occur. Although this means for an 'arbitrary' SEM model we have identifiability, this does not rule out well-known cases including linear+Gaussian where the existence of an equivalent backwards model is unavoidable.

**Remark 2.** Second, this result is applied locally to a single child node, rather than the global SCM structure. Previous work has partially explored the global structure in the population limit, see Chickering [2002] and Peters et al. [2014]; however, a priori, the constrained solutions space may grow exponentially large, leading to practical limitations in the real-world setting with finite samples.

We next provide the relevant extension to recover the hypergraphical structure as well. Indeed, if the DAG structure is identifiable (at least locally), then the hyper DAG structure is also identifiable (at least locally). This also implies that when the entire DAG structure is identifiable in the ANM case, the entire hyper DAG structure is also identifiable.

**Theorem 3.** Suppose we have two forward models given by two alternate collections $\mathcal{I} \subseteq \mathcal{P}([d])$ and $\mathcal{J} \subseteq \mathcal{P}([d])$, where $\mathcal{P}$ denotes the power set, with models:

$$p(y|x_1, \ldots, x_d) = p_n(y - \sum_{S \in \mathcal{I}} f_S(x_S)) \qquad p(y|x_1, \ldots, x_d) = p_{\tilde{n}}(y - \sum_{T \in \mathcal{J}} g_T(x_T)) \quad (16)$$

Assume that in addition to the assumptions of Theorem 2, we also have that the functions $f_S$ and $g_T$ are differentiable up to order $\max\{\max_{S \in \mathcal{I}}\{|S|\}, \max_{T \in \mathcal{J}}\{|T|\}\}$, or more simply up to order $d$. It then follows that $\mathcal{I} = \mathcal{J}$ and thus the hypergraph structures of the two models are exactly the same. Moreover, outside of trivial modifications to the $f$'s and $g$'s, the two functional generating models are exactly the same.

Finally, we restrict further to the case of Gaussian noise variables in the ANM to directly recover a global hypergraphical result, further relating Sections 2.2 and 2.3.

**Theorem 4.** Suppose that all additive noise variables are drawn from a Gaussian distribution, implying that $\nu(\varepsilon) = \frac{-1}{2\sigma^2}\varepsilon^2$ (or some general quadratic form) for all $i \in [d]$. Suppose also that we generate data according to the directed hypergraph $\mathcal{H}$. The undirected, immoralized version of this hypergraph $\mathcal{H}'$ is identifiable directly from the data in the sense that $\xi(x_1, \ldots, x_d) = \sum_{S \in \mathcal{H}'} \xi_S(x_S)$ for some arbitrary functions $\xi_S$. In other words, the undirected hypergraph from setting 1 is directly identifiable from the distribution and thus settings 2 and 3 overlap for ANMs with Gaussian noise.

All proofs may be found in the appendix.

# 4 Algorithm

Our method for hypergraph discovery and structural equation modeling heavily entwines the schema of CAM [Bühlmann et al., 2014] and the higher-order interaction techniques of SIAN [Enouen and Liu, 2022]. Accordingly, we review both algorithms in much greater detail in Appendix B. Here in the main body of the paper, we briefly review the three step procedure introduced by CAM and discuss our HCAM extension to their original approach. The first stage is a preselection phase which searches for good candidate parents for each potential child node. The second stage is the bulk of the algorithm, greedily constructing the DAG via including each directed edge one at a time based on the improvement to the log-likelihood. The third stage is a final pruning stage which does not change the topological order, but simply removes parents which are no longer thought to be relevant.

## 4.1 Step 1: Candidate Search

The first stage of CAM Bühlmann et al. [2014] finds candidate edges/ parents by training a GAM regression on each of the $d$ variables. In their work, this was mainly necessary for them in the high-dimensional setting, where explicit consideration of so $\mathcal{O}(d^2)$ edges when $d$ is large poses a challenge. However, for HCAM, this stage becomes absolutely critical. That is because we not only need to look at all $(d^2 - d)$ candidate directed edges of CAM, but also all the $\frac{1}{2}(d^3 - 3d^2 + 2d)$

candidate directed tri-edges, as well as higher-order hyperedges, etc. To consider all hyperedges directly without any heuristic would require considering an exponential number of hyperedges.

Accordingly, we first use a deep neural network to 'search' for good hyperedges, following steps 1 and 2 of SIAN [Enouen and Liu, 2022]. That is, for each of the $d$ variables, we regress an MLP DNN to minimize the mean-squared error (Gaussian likelihood). We then use an XAI technique, called Archipelago [Tsang et al., 2020], to give an importance score for each of the feature interactions involving the other variables.

## 4.2 Step 2: Greedy Selection

The next phase consists of the bulk of the algorithm and can also be considered the most important part and yet most simplistic part. A greedy heuristic is taken to gradually build the DAG from the initialized empty set of vertices. For each of the possible edges (or potentially the subset selected in step 1), a new likelihood model is trained to simulate adding the edge to the DAG. Importantly, because of the independence of the ANM noise, this is simply measured as the drop in MSE with and without the additive term. Gradually, edges are added until some stopping point and the final phase of pruning begins.

In our case, several small adjustments must be made to deal with the case of HDAGs. Importantly, unlike CAM, HCAM must keep track of both the HDAG and the induced partial-order matrix simultaneously. Generically, we follow the exact same procedure, training higher-order additive models with each candidate hyperedge to see the improvement in MSE. We start with 10 candidates for each of the $d$ variables, based on the ranking provided in step 1, and we replenish the candidates if there are ever less than 5 viable hyperedges per a variable. This cutdown is able to reduce the number of higher-order SIAN additive models we train, which helps in improving the overall runtime.

## 4.3 Step 3: Final Pruning

Finally, the full model is trained end-to-end once more with all of the included edges from step 2. The final stage simply removes edges corresponding to additive terms which are too close to zero, and thus likely to be useless in the causal model. Our extensions prunes in the exact same way, with higher-order terms corresponding to higher-order edges, but there is no practical difficulty in doing this extension. The major difficulty of this part is choosing a threshold which corresponds with a nuisance parameter. The original CAM work uses a threshold of $0.001$ for the p-values provided alongside the GAM models. We use neural networks which do not provide p-values and thus somewhat similarly threshold based on the MSE of the additive term, using a threshold of $1.0e{-}4$.

# 5 Experiments

We compare across multiple synthetic datasets obeying the additive noise model (ANM) while varying the degree of the additive models. Following previous works, we generate the base DAG from an Erdos-Renyi random graph with an average of 4 edges per node. We generate 1D, 2D, and 3D additive models to distinguish different hypergraphical structures. For 1D models, we avoid the linear model to allow for identifiability and use a random Gaussian process to define the additive functions. For 2D and 3D models, we use multilinear terms, $\beta_{jk}x_jx_k$, with coefficients drawn around $\pm1$, then normalizing by the total coefficient weight for a parent set. We assume the additive noise terms are coming from a Gaussian distribution and draw random variances constrained to be close to $1.0$. We set $d = 30$ (number of nodes) and $n = 10000$ (number of samples) in our experiments.

We compare against baselines of PC [Spirtes and Glymour, 1991], GES [Chickering, 2002], BOSS [Andrews et al., 2023], RESIT [Peters et al., 2014], CAM [Bühlmann et al., 2014], and SCORE [Rolland et al., 2022]. We additionally compare against a baseline which assumes the empty graph, equivalent to assuming all of the observed variables are completely indepedent. We compare against the structural Hamming distance (SHD) and the structural interventional distance (SID) obtained by each method on different complexities of synthetic data. We additionally compare the Hamming distance on the hypergraph bodies, based on the assumption in Figure 1f, calling it the higher-order structural Hamming distance. Because the maximal error is on the order $2^{30} \approx$ one billion, in some cases, we do not compute exactly and report a lower bound in Table 4.

Table 2: Structural Hamming Distance for ER4 and N=10,000.

| | BOSS | CAM | GES | PC | SCORE | RESIT | zero | HCAM |
|---|---|---|---|---|---|---|---|---|
| ER4 1D | 189.67±20.24 | **42.67±4.50** | 206.67±11.90 | 159.67±21.17 | **69.33±20.24** | 494.67±3.30 | 128.67±8.38 | **67.67±13.82** |
| ER4 2D | 115.33± 1.70 | 134.67±2.49 | 116.00± 2.16 | 134.00± 4.32 | 126.00± 2.83 | 146.33±5.79 | **115.33±1.70** | **107.00± 0.82** |
| ER4 3D | 109.00± 4.24 | 120.33±1.89 | 106.67± 3.86 | 120.33± 3.86 | 117.00± 5.89 | 132.33±6.34 | **106.67±3.86** | 106.67± 3.86 |

Table 3: Structural Intervention Distance for ER4 and N=10,000.

| | BOSS | CAM | GES | PC | SCORE | RESIT | zero | HCAM |
|---|---|---|---|---|---|---|---|---|
| ER4 1D | 608.33±45.09 | **0.00± 0.00** | 646.67±54.97 | 748.33±66.04 | **105.33±47.12** | 646.67±54.97 | 726.00±64.19 | **525.33±64.25** |
| ER4 2D | 679.33±38.94 | 750.67± 7.93 | 687.33±39.67 | 713.67±60.18 | 745.67±16.11 | 734.67±21.64 | **679.33±38.94** | **661.00±31.03** |
| ER4 3D | **647.67±15.08** | 744.67±22.23 | 679.00±39.02 | 699.00±46.31 | 751.67±24.14 | 744.00±29.44 | **679.00±39.02** | 679.00±39.02 |

Table 4: Higher-Order Structural Hamming Distance for ER4 and N=10,000.

| | BOSS | CAM | GES | PC | SCORE | RESIT | zero | HCAM |
|---|---|---|---|---|---|---|---|---|
| ER4 1D | >10,000 | **42.67± 4.50** | >10,000 | >1,000 | >10,000 | >10,000 | 128.67± 8.38 | **48.33± 8.18** |
| ER4 2D | 157.33± 1.70 | 168.00± 5.35 | 158.00± 1.41 | 183.67±10.78 | 171.33± 2.05 | 602.33±603.66 | 157.33± 1.70 | **119.00± 5.35** |
| ER4 3D | 236.00±18.38 | 256.33±14.66 | 248.67±18.15 | 264.67±23.23 | 264.00±14.76 | 263.00± 13.44 | **248.67±18.15** | 248.67±18.15 |

## 5.1 Results

Overall, we find that many methods are successful for the simpler 1D data obeying the CAM assumptions, especially the CAM and SCORE algorithms. HCAM does not achieve the same level of success as these algorithms but still achieves good performance on this dataset. Surprisingly, all other methods have rather great difficulty in identifying the causal structure.

On our specifically higher-order datasets, however, we find that the story is quite different. In particular, the only algorithm which is able to defeat the baseline on the 2D data is our HCAM method specifically focusing on modeling the 2D interactions. In the 3D data, none of the algorithms we run are able to find empirical success over the zero baseline. That is to say, we should have just assumed the variables were independent and not run our algorithm at all.

This lack of capability is despite the fact that we used a simple DGP (multilinear plus Gaussian) on a relatively low number of variables ($d = 30$) and provided a relatively standard number of observations ($n = 10,000$). Aligning with our hypothesis that the statistical complexity increases in the presence of higher-order interactions, we strongly believe this points to some aspects of structure discovery research which have received less attention but remain highly influenced by the presence of interactions.

## 6 Conclusion

We have introduced a framework for considering the impact of higher-order interactions on causal structure. After introducing the relevant definitions, we further show the identifiability of the introduced structure across multiple cases of interest. Finally, we demonstrate the potential usefulness of the hypergraphical structure in empirical case using the additive noise model, along with providing a first algorithm for adequately handling the higher-order structure directly from observed data. This perspective additionally allowed us to identify a potential blindspot of many existing structure discovery approaches: their lack of focus on statistical power and lesser ability to handle higher-order interactions.

We envision future work may continue to benefit from a perspective using higher-order interactions. Some directions of future exploration include improving on the algorithms and theoretical results presented herein, potentially solving increasingly challenging datasets constructed using the higher-order perspective on the generated variables. Extension to appropriately handling latent variables and latent confounding is a direction of serious interest. Hypergraphical structures being identifiable from the data distribution alone, extending existing MEC and identifiability results, point to the potential of hypergraphical structure across even more contexts and settings than the ones explored herein.

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

# A  Proofs of Theorems

## A.1  Proof of Theorem 1

To begin, we remind that while the classical MEC formulation is concerned with mapping the conditional independencies of a distribution with the Markov conditions and structure of a DAG, we are here concerned with mapping the conditional multi-independencies of a distribution with the Markov conditions and structure of an HDAG. Accordingly, it is first worth noting that because the conditional multi-independencies of a distribution is a strictly larger set of conditions that the original set of conditions, hence, we get the existing result on the DAG corresponding to the HDAG, let's call it the reduced DAG. This is done in the obvious way by taking the union of all a node's hyperparents and defining them as the parent set.

Thus, in addition to the skeletons and (unshielded) colliders of the DAG which are already identifiable from the typical conditions, we must investigate which HDAGs are further distinguishable via these conditions and which HDAGs are distinguishable via the new conditions.

We will start with the easier point about the body of an HDAG. Once again, this is defined by removing all directed arrows from the directed hypergraph. In the same way that a pair of nodes $i$ and $j$ are 'inseparable' if they are not conditionally indepedent for any conditioning set, we can say that a triple of nodes $i$, $j$, and $k$ are inseparable if they are not conditionally multi-independent for any conditioning set. In much the same way this indicates the existence of an edge in the DAG case, this will indicate the existence of a (three-dimensional) hyperedge in the HDAG case.

It is thus straightforward to see that the existence of an inseparable triple shows the existence of a directed hyperedge (where one of the three vertices is the child). Further, this generalizes to all degrees in the exact same way. It is briefly reminded that the hierarchy constraint plays a role of convenience here in the sense that a higher-order edge is detected via a three-dimensional hyperedge even if, say, the generating equations do not make this explicit. This exactly parallels what happens in the 2D case with an inseparable pair of nodes, where the DAG edges are capturing everything 'between $i$ and $j$ or higher'. To be explicit, the DAG edge $i \rightarrow j$ could have a second parent of $j$ which interacts with $i$ when generating $j$. Nonetheless, it is clear from these conditions that we may directly identify the body of the HDAG, and that the body of the HDAG is strictly more informative than the skeleton of the HDAG's reduced DAG.

Now let us move on to the discussion of colliders between parents. To prepare for our generalization of colliders, we first allude to the fact that in Equation 7, we can see directly that the normalizing $\mathcal{Z}$ score over the parent set is the cause of a collider. In particular, unlike the natural $\theta$ terms which cannot be destroyed via marginalization of variables, the $\mathcal{Z}$ terms are destructible under marginalization of the child. This naturally corresponds to the more typical conditions noting that there is some smaller set (not including the child) where conditioning provides independence but conditioning on the child additionally breaks the independence. Of course, not all sets without the child included is able to marginalize out the child, namely, conditioning on any descendant of the child is also problematic.

Nonetheless, we may proceed by extending the definition in the same way. We say that a set of nodes $i$, $j$, $k$ alongside a fourth node $\ell$ are a tricollider so long as there exists some conditioning set $S$ under which the $i$, $j$, $k$ are not conditionally tri-dependent; however, after additionally conditioning on the node $\ell$ (their joint child), the tri-independence breaks and $i$, $j$, $k$ are conditionally tri-dependent when conditioning on $(S + \{\ell\})$. Equally, it can be seen that $i$, $j$, $k$ are the joint parents of $\ell$ whose $\mathcal{Z}$ normalization appears only when needing to condition on $\ell$.

In fact, it is now extremely straightforward to state our faithfulness condition directly. We say that a distribution observes a hypergraph structure faithfully so long as no natural theta term is destructible via marginalization; moreover, the joining of two theta terms via marginalization will lead to a new theta term with the maximal relative size (i.e. no higher-order terms magically cancel and zero out).

Finally, because of this faithfulness condition, we can equally start with the log-probability score function which obeys the Hyper Markov property and construct all possible multi-dependence tests in the backwards direction. Accordingly, no $\theta$ term will cancel via marginalization and each $\mathcal{Z}$ will only cancel via marginalization of its respective child. As a reminder, conditioning on a set is easier via implicitly pulling out the variables value in the conditional distribution, and marginalizing has been made possible via the faithfulness distribution. Together, these allow us to describe all

of the energy terms of a new conditioned distribution and one can directly read off the conditional multi-dependence via the existence of or lack of the highest-order energy term.

## A.2 Proof of Theorem 2

*Proof.* The arguments here closely follow the original arguments for Theorem 1 of Hoyer et al. [2008].

First, recall that we will write

$$
\begin{aligned}
\pi(x_{[d]}, y) &= \nu(y - f(x)) + \xi(x) \\
\pi(x_{[d]}, y) &= \tilde{\nu}(x_i - g(x_{-i}, y)) + \eta(x_{-i}, y)
\end{aligned}
$$

(17)

We may first proceed with some basic calculations

$$
\begin{aligned}
\frac{\partial}{\partial x_i}\pi &= \nu' \cdot -\frac{\partial f}{\partial x_i} + \frac{\partial \xi}{\partial x_i} \\
&= \tilde{\nu}' \cdot 1 + 0 \\
\frac{\partial}{\partial y}\pi &= \nu' \cdot 1 + 0 \\
&= \tilde{\nu}' \cdot -\frac{\partial g}{\partial y} + \frac{\partial \eta}{\partial y}
\end{aligned}
$$

And further

$$
\begin{aligned}
\frac{\partial^2}{\partial x_i^2}\pi &= \nu'' \cdot \frac{\partial f}{\partial x_i} \cdot \frac{\partial f}{\partial x_i} - \nu' \cdot \frac{\partial^2 f}{\partial x_i^2} + \frac{\partial^2 \xi}{\partial x_i^2} \\
&= \tilde{\nu}'' \\
\frac{\partial^2}{\partial x_i \partial y}\pi &= \nu'' \cdot -\frac{\partial f}{\partial x_i} \\
&= \tilde{\nu}'' \cdot 1 \cdot -\frac{\partial g}{\partial y} + \tilde{\nu}' \cdot 0 + 0 = -\tilde{\nu}'' \cdot \frac{\partial g}{\partial y}
\end{aligned}
$$

So it follows from the $\tilde{\nu}$ equation that

$$
\frac{\frac{\partial^2 \pi}{\partial x_i^2}}{\frac{\partial^2 \pi}{\partial x_i \partial y}} = \frac{\tilde{\nu}''}{-\tilde{\nu}'' \cdot \frac{\partial g}{\partial y}} = \frac{-1}{\frac{\partial g}{\partial y}}
$$

And further

$$
\frac{\partial}{\partial x_i}\left[\frac{\frac{\partial^2 \pi}{\partial x_i^2}}{\frac{\partial^2 \pi}{\partial x_i \partial y}}\right] = \frac{\partial}{\partial x_i}\left[\frac{-1}{\frac{\partial g}{\partial y}}\right] = 0
$$

Plugging this back in to the equation with $\nu$ gives us

$$
\frac{\partial}{\partial x_i}\left[\frac{\nu'' \cdot f' \cdot f' - \nu' f'' + \xi''}{-\nu'' f'}\right] \equiv 0
$$

Application of repeated derivative rules and simplification gives the required

$$
\xi''' = \xi'' \cdot \left(-\frac{\nu''' f'}{\nu''} + \frac{f''}{f'}\right) +
$$
$$
\left(-2\nu'' f'' f' + \nu' f''' + \frac{\nu' \nu''' f' f''}{\nu''} - \frac{\nu' f'' f''}{f'}\right)
$$

$\square$

## A.3 Proof of Theorem 3

*Proof.* Supposing that we have two different forward models given by

$$p(x_1, \ldots, x_d, y) = p_n(y - \sum_{S \in \mathcal{I}} f_S(x_S)) \cdot p_x(x_1, \ldots, x_d) = p_n(y - f(x)) \cdot p_x(x_1, \ldots, x_d)$$

$$p(x_1, \ldots, x_d, y) = p_{\tilde{n}}(y - \sum_{T \in \mathcal{J}} g_T(x_T)) \cdot p_x(x_1, \ldots, x_d) = p_{\tilde{n}}(y - g(x)) \cdot p_x(x_1, \ldots, x_d)$$

We may take $\pi$ as before and see that

$$\frac{\partial}{\partial y}\pi = \qquad\qquad\qquad \nu' = \qquad\qquad\qquad \tilde{\nu}'$$

$$\frac{\partial}{\partial x_i}\pi = \quad -\nu' \cdot \Big[ \sum_{S \in \mathcal{I}} \frac{\partial f_S}{\partial x_i}(x_S)\mathbb{1}(i \in S) \Big] = \quad -\tilde{\nu}' \cdot \Big[ \sum_{T \in \mathcal{J}} \frac{\partial g_T}{\partial x_i}(x_T)\mathbb{1}(i \in T) \Big]$$

It follows as before that we may write

$$-\Big[ \frac{\frac{\partial}{\partial x_i}\pi}{\frac{\partial}{\partial y}\pi} \Big] = \quad \Big[ \sum_{S \in \mathcal{I}} \frac{\partial f_S}{\partial x_i}(x_S)\mathbb{1}(i \in S) \Big] = \quad \Big[ \sum_{T \in \mathcal{J}} \frac{\partial g_T}{\partial x_i}(x_T)\mathbb{1}(i \in T) \Big]$$

Further, we have that

$$0 \equiv \frac{\partial}{\partial x_{R-i}}\Big[0\Big] = \frac{\partial}{\partial x_{R-i}}\Big[ \frac{\frac{\partial}{\partial x_i}\pi}{\frac{\partial}{\partial y}\pi} - \frac{\frac{\partial}{\partial x_i}\pi}{\frac{\partial}{\partial y}\pi} \Big] = \frac{\partial}{\partial x_{R-i}}\Big[ -\frac{\partial f}{\partial x_i} + \frac{\partial g}{\partial x_i} \Big] = -\frac{\partial f}{\partial x_R} + \frac{\partial g}{\partial x_R}$$

$$= -\Big[ \sum_{S \in \mathcal{I}} \frac{\partial f_S}{\partial x_R}(x_S) \cdot \mathbb{1}(R \subseteq S) \Big] + \Big[ \sum_{T \in \mathcal{J}} \frac{\partial g_T}{\partial x_R}(x_T) \cdot \mathbb{1}(R \subseteq T) \Big] \quad (18)$$

Recall that we assumed that $\mathcal{I}$ and $\mathcal{J}$ are downwards closed or 'hierarchical' which means all $S \in \mathcal{I}$ have all its subsets $S' \subseteq S$ also inside of $\mathcal{I}$. If we then take $R \notin \mathcal{I}$, it implies that $R \not\subseteq S$ for all $S \in \mathcal{I}$, otherwise such an $S$ would not obey the downwards closed property. This means that $\mathbb{1}(R \subseteq S) = 0$ and

$$0 \equiv -0 + \Big[ \sum_{T \in \mathcal{J}} \frac{\partial g_T}{\partial x_R}(x_T) \cdot \mathbb{1}(R \subseteq T) \Big]$$

This means that for all $T \in \mathcal{J} \cap \mathrm{up}(R)$ where $\mathrm{up}(R) := \{T : T \supseteq R\}$ it must be the case that the $R$-th partial derivative is zero. We will focus on the case of $T = R$, but the same holds for all $T$ as above.

We may take a modification of the function $g_T$ such that it is instead represented by additive functions of a lower degree. This is equivalent to saying that $g_T \equiv 0$ and hence $T \in \mathcal{J}$ was actually a contradiction. Let us see that $\frac{\partial g_R}{\partial x_R} \equiv 0$, so then writing $R = \{r_1, \ldots, r_{|R|}\}$, we have that $\int_{x_{r_1}} \frac{\partial g_R}{\partial x_R} = C_1(x_{r_2}, \ldots, x_{r_{|R|}})$ for some function $C_1$ which is constant with respect to $x_{r_1}$. Further $\int_{x_{r_2}} \frac{\partial g_R}{\partial x_{R-r_1}} = \int_{x_{r_2}} C_1(x_{r_2}, \ldots, x_{r_{|R|}}) = C_1(x_{r_2}, \ldots, x_{r_{|R|}}) + C_2(x_{r_1}, x_{r_3}, \ldots, x_{r_{|R|}})$ and continuing on, we may ultimately see that $g_R = \sum_{R' \subsetneq R} g_{R'}$ which means by our assumption that $\mathcal{J}$ is a minimal representation with no zero additive models, that actually $T \notin \mathcal{J}$.

The same arguments may be taken in reverse to show that for all $R \notin \mathcal{J}$, it must be the case $R \notin \mathcal{I}$. There is a mild difference in the way the contradiction is applied argument when reversing the arguments because we take the perspective that $\mathcal{I}$ is the ground truth generating process and $\mathcal{J}$ is a potential alternate model (i.e. the contradiction is on $R$ not on $S$.) Nonetheless, this results in the fact $\mathcal{I} = \mathcal{J}$ for any two forward models which are representing the same distribution, and that the functional representations are moreover the same. The latter part can be seen directly from $\frac{\partial f}{\partial x_i} = \frac{\partial g}{\partial x_i}$ for all $i \in [d]$ and similar arguments for removing trivial terms from the additive model. The final modification which may exist is up to a constant, which is resolved by the differences in the mean of the variable represented by $\nu$ and $\tilde{\nu}$. This is solved by the ANM assumption that the additive noises are mean-centered. $\qquad\square$

Altogether, this is taken to mean that whenever the DAG is locally identifiable (and thus there are only valid forward models and no potential backwards model), then the hypergraph structure of the forward model is additionally identifiable. This additionally implies that if the entirety of the DAG is identifiable, then the entirity of the hyper DAG is also identifiable.

## A.4 Proof of Theorem 4

*Proof.* Under the further assumption that $\varepsilon_i \sim \mathcal{N}(0, \sigma_i^2)$, we have that $\nu(\varepsilon) = \log p(\varepsilon) = \log\left(\frac{1}{\sqrt{2\pi\sigma^2}} \cdot \exp(-\frac{\varepsilon^2}{2\sigma^2})\right) = -\log(2\pi\sigma^2) - \frac{1}{2\sigma^2}\varepsilon^2$. Further, we have that $\nu'(\varepsilon) = -\frac{1}{\sigma^2}\varepsilon$, $\nu''(\varepsilon) = -\frac{1}{\sigma^2}$, and $\nu^{(k)}(\varepsilon) = 0$ for larger k. Accordingly, we may write the entire distribution as

$$\xi(x_1, \ldots, x_d) = \log p(x_1, \ldots, x_d) = \sum_i \log p(x_i | x_{\mathrm{Pa}(i)})$$

$$= \sum_i \nu_i(\varepsilon_i) = \sum_i \nu_i\left(x_i - \sum_{(S,i)\in\mathcal{H}} f_{S\to i}(x_S)\right)$$

Let us write $f_{\to i}$ to denote $\sum_{(S,i)\in\mathcal{H}} f_{S\to i}$ and $F_i(x) = (x_i - f_{\to i}(x))$. It is straightforward to verify through repeated applications of the chain rule and product rule that

$$\frac{\partial^n}{\partial x_1, \ldots, \partial x_n} \nu_i\Big(F_i(x)\Big) = \nu_i \cdot \frac{\partial^n}{\partial x_1, \ldots, \partial x_n}\Big(F_i(x)\Big) + \nu_i' \cdot \sum_{\emptyset \subsetneq A \subsetneq [n]} \frac{\partial^{|A|}}{\partial x_A}\Big(F_i(x)\Big) \cdot \frac{\partial^{n-|A|}}{\partial x_{[n]-A}}\Big(F_i(x)\Big)$$

It can further be seen that

$$\frac{\partial^{|A|}}{\partial x_A}\Big(F_i(x)\Big) = \frac{\partial^{|A|}}{\partial x_A}\Big(x_i - f_{\to i}(x)\Big) \tag{19}$$

is zero whenever $A$ is not all $i$'s and $A$ is not a subset of one of the $S$ where $(S, i) \in \mathcal{H}$. This means exactly that

$$\sum_{\emptyset \subsetneq A \subsetneq [n]} \nu_i' \cdot \frac{\partial^{|A|}}{\partial x_A}\Big(F_i(x)\Big) \cdot \frac{\partial^{n-|A|}}{\partial x_{[n]-A}}\Big(F_i(x)\Big)$$

is barely nonzero whenever we take $[n]$ equal to $(S + i)$ for some $(S, i) \in \mathcal{H}$. Note that this is equivalent to taking some $(S + i) \in \mathcal{H}'$ where we recall $\mathcal{H}'$ is the undirected version of the directed graph $\mathcal{H}$. If we instead take some $R \notin \mathcal{H}'$, then it will be the case that this derivative is zero, because all $A \subseteq R$ will have either $\frac{\partial^{|A|}}{\partial x_A} \equiv 0$ or $\frac{\partial^{n-|A|}}{\partial x_{[n]-A}} \equiv 0$. Moreover, it is the case that the first term is clearly zero $\frac{\partial^{|R|}}{\partial x_R}\Big(F_i(x)\Big) \equiv 0$.

Since this is true for all $i$ so long as we are taking $R \notin \mathcal{H}'$, we have that

$$\frac{\partial^{|R|}}{\partial x_R}\xi(x_1, \ldots, x_d) = \frac{\partial^{|R|}}{\partial x_R} \sum_i \nu_i\left(x_i - \sum_{(S,i)\in\mathcal{H}} f_{S\to i}(x_S)\right) \equiv 0$$

Following the same approach as in the proof of Theorem 3, this means that we are able to write $\xi(x_1, \ldots, x_d) = \sum_{S\in\mathcal{H}'} \xi_S(x_S)$ for some functions $\xi_S$.

$\square$

Note that the decomposition of the likelihood function's structure does not contradict existing results saying that the directionality of the graphical model is not always identifiable from data. In particular, in the linear-Gaussian case, it may not be possible to distinguish which direction is the causal direction. Nonetheless, the graphical structure which is recovered in this purely Gaussian case corresponds to what is available in the precision matrix [Loh and Bühlmann, 2014], still identifying the undirected graphical structure underlying the distribution. Under the CAM-like assumptions of linear SEMs, the hypergraph structure is reduced to its simplest representation, which is isomorphic to a graphical representation.

# B Full Algorithm Details

## B.1 Causal Additive Model

CAM (Causal Additive Model) uses a three step procedure to discover a set of additive structural equations according to Equation **??**. First, a preliminary search is made over the directed edges using sparse regression to cut down on the search space. Second, a greedy algorithm gradually adds the best edges to the DAG so long as it does not create any cycles. Third, the final DAG structure's additive models are trained once again with sparse regression to encourage the removal of extraneous edges.

**Step 1** First, a preliminary search is made over all possible edges via sparse regression. For each variable $j \in [d]$, one fits an additive model based on all of the other possible directed edges $(k, j)$ using sparse regression. This allows for a smaller subset of the quadratic number of edges to be considered, especially in the high-dimensional setting when $d$ is large.

The mean-squared error objective is minimized based on the assumption that the noise terms are Gaussian.

$$\log p(\varepsilon_j) = -\log(2\pi\sigma_j^2) - \frac{1}{2\sigma_j^2} \cdot \varepsilon_j^2 \tag{20}$$

$$\hat{\sigma}_j^2 := \|X_j - \hat{X}_j\|^2 = \|X_j - \sum_k \hat{f}_{k \to j}(X_k)\|^2 \tag{21}$$

**Step 2** Second, the bulk of the algorithm centers around a greedy approach for gradually adding directed edges which do not disagree with the partial structure which has been built up so far. Every edge from the local neighborhood determined in step 1 is considered to be added, so long as it would not create a cycle in the DAG. Each edge is ranked by its ability to improve the log-likelihood of the overall model, by training an additive model with the selected edges.

$$\hat{\sigma}_j^2(\mathcal{N}_j) := \|X_j - \sum_{k \in \mathcal{N}_j} \hat{f}_{k \to j}(X_k)\|^2 \tag{22}$$

$$(k*, j*) = \underset{(k,j) \quad \text{acyclic}}{\operatorname{argmin}} \left\{ \hat{\sigma}_j^2(\mathcal{N}_j \cup \{k\}) - \hat{\sigma}_j^2(\mathcal{N}_j) \right\} \tag{23}$$

Importantly, for $j \neq j*$, it is not necessary to retrain the additive models to recompute the values of $\hat{\sigma}_j^2(\mathcal{N}_j)$, because they are not affected by the inclusion of edges in other parts of the graph (except that it may block an edge from being added due to the acyclicity constraint).

**Step 3** Lastly, the collection of directed edges which were selected in step 2 are used to train a final model end-to-end, with additional regularization designed to shrink unnecessary edges to become sparse. Additive terms $\hat{f}_{k \to j}$ which are deemed insignificant are removed from the model completely and the final set of edges define the final DAG.

## B.2 Sparse Interaction Additive Network

SIAN (Sparse Interaction Additive Network) is an approach designed to train higher-order additive models using neural network techniques. This approach also consists of three main phases. In our case, we will follow CAM's implicit Gaussian assumption by minimizing the mean-squared error objective which corresponds with the likelihood of independent Gaussian variables.

In the first phase, a typical neural network $f_\theta$ is trained to predict an output variable in terms of the input variables.

$$\hat{\sigma}^2(\theta) := \|Y - \hat{Y}(\theta)\|^2 = \|Y - \hat{f}_\theta(X_{[d]})\|^2 \tag{24}$$

In the second phase, interpretability techniques are combined with a special feature interaction selection (FIS) algorithm which ensures a sufficient coverage of the complex space of interactions while avoiding the exponential blow up in complexity from exploring all higher-order interactions. The final result of the first two phases is a collection $\mathcal{I} \subseteq \mathcal{P}([d])$ which is some collection of all of the feature interactions $S$ which are important to predicting the output variable. Finally, the set of collected higher-order interactions are then used to train a neural-network-based additive model which obeys the interaction structure determined in the selection algorithm.

$$\hat{\sigma}_{\mathcal{I}}^2(\theta) := \|Y - \hat{Y}_{\mathcal{I}}(\theta)\|^2 = \|Y - \sum_S \hat{f}_{S,\theta_S}(X_S)\|^2 \tag{25}$$

This final additive neural network has pleasant properties like being more interpretable as well as more robust than the original neural network. In our context, we will use these neural additive models as the major component of modeling the hypergraphical additive structure we assumed previously.

### B.3 Higher-order Causal Additive Model

In our algorithm, we broadly follow the same steps as the original CAM algorithm, replacing all components which are limited to one-dimensional additive models with their higher-order counterparts.

In the first step of our algorithm, we must reduce the number of candidate edges which will be considered in the downstream steps. Although CAM mentions this is only necessary in the high-dimensional setting for their additive assumption, ours is absolutely necessary except in extremely small cases (perhaps $d \leq 5$). This is because instead of searching over all candidate directed edges, $\{(k, j) : k \neq j, j \in [d]\}$, we must perform a search over the much larger space of all candidate directed hyperedges, $\{(S, j) : S \subseteq ([d] \setminus j), j \in [d]\}$.

For this purpose, we employ the first two phases of SIAN to each of the variables. That is, for each $j \in [d]$, we train a neural network to predict $X_j$ from $X_{-j}$ and then run a feature interaction selection algorithm to find a neighborhood of important interactions which are useful for predicting $X_j$.

$$\hat{\sigma}_j^2(\theta) := \|X_j - \hat{X}_j(\theta)\|^2 = \|X_j - \hat{f}_{j,\theta}(X_{[d]-k})\|^2 \tag{26}$$

$$\mathcal{I}_j = \text{FeatureInteractionSelection}(\hat{f}_{j,\theta}) \tag{27}$$

These selected interactions are then taken as the candidate set of directed hyperedges to be used in the later parts of the algorithm.

$$\tilde{\mathcal{H}} := \{(S, j) : S \in \mathcal{I}_j, j \in [d]\} \tag{28}$$

Note that these hyperedges are also given an importance score from the original FIS algorithm and may be sorted by their a priori importance.

In the second step of our algorithm, we follow the greedy approach of including hyperedges based on the improvement to the log-likelihood. This requires training many different additive models obeying the interaction constraints imposed by the current hyper DAG. We use the additive models from the third phase of SIAN to minimize an MSE objective as before.

In particular, we train multiple SIAN additive models to compare the different improvements in scores coming from adding all possible hyperedges $(S, j) \in \tilde{\mathcal{H}}$. This improvement in score is again interpreted as the improvement in reducing the noise of the added Gaussian via reduction in MSE.

$$\hat{\sigma}_j^2(\mathcal{N}_j') := \|X_j - \sum_{S \in \mathcal{N}_j'} \hat{f}_{S \rightarrow j}(X_S)\|^2 \tag{29}$$

$$(S*, j*) = \underset{(S,j) \quad \text{acyclic}}{\text{argmin}} \left\{ \hat{\sigma}_j^2(\mathcal{N}_j' \cup S) - \hat{\sigma}_j^2(\mathcal{N}_j') \right\} \tag{30}$$

Because this requires training a large amount of additive models, we make multiple concessions to allow for a more rapid selection process during step 2 which can otherwise take a significant chunk of the overall algorithm time. Because of our higher-order neighborhood selection from step 1, it is at least feasible to search over higher-order interactions without facing an exponential blowup in the number of additive models which must be trained.

However, in practice we further reduce the number of additive models we train to a maximum of 10 interactions per each variable $X_j$. As tuples are selected from the candidate superset $\tilde{\mathcal{H}}$ to be actually included into the model, additional candidate hyperedges are replenished to be explored in future iterations of the step 2 loop.

Furthermore, instead of training these SIAN additive models until there is no further reduction in MSE, we only train each for five epochs in total. We find that this gives a strong enough measurement of the performances of the differnt additive models without cutting significantly into the overall time taken. Moreover, because the heuristic coming from the first two phases of SIAN used in step 1 of our algorithm is generally quite good, an approximate measure of the reduction in score from each hyperedge in step 2 seems to generally be sufficient.

In the third step of our algorithm, we again follow the CAM setup and train an end-to-end SIAN model which obeys the structure which was greedily added in step two of the algorithm. L1 regularization terms are used on each of the shape functions in the additive model to encourage shrinkage in the unnecessary terms of the additive model. Similar to CAM, unimportant additive terms are thresholded away and removed from the final hyper DAG.

$$\mathcal{L}(\theta) := \tag{31}$$
$$\sum_j \|X_j - \sum_{(S,j)\in\mathcal{H}} \hat{f}_{S\to j}\|^2 + \lambda_1 \sum_{(S,j)\in\mathcal{H}} |\hat{f}_{S\to j}|$$

For comparison against other DAG-based methods, the hyper DAG is further projected onto its equivalent DAG formulation.

