# OpenReview forum: "Higher-Order Causal Structure Learning with Additive Models"
_NeurIPS.cc/2025/Conference — Submitted to NeurIPS 2025_

### Official Review · Reviewer_nSx3 · 2025-06-10

**Clarity:** 3
**Significance:** 2
**Originality:** 3
**Rating:** 3
**Confidence:** 3

**Summary:**

The paper studies generalizing causal modelling to hypergraphs with the goal of developing more capable methods for dealing with higher-order variable interactions. In addition to showing three hypergraph generalizations (Markov networks, Bayesian networks, additive noise models), the authors design a CAM-inspired algorithm for learning a hyper DAG and assess the performance of their algorithm.

**Questions:**

I would appreciate it if the authors could address my concerns about the scope of the experiments in their rebuttal.

**Ethical Concerns:**

["NO or VERY MINOR ethics concerns only"]

**Final Justification:**

While I appreciate how active the Authors were during the rebuttal phase, my main concern (limited real-world significance; showable with e.g. experiments based on real-world instances) unfortunately remains.

As I mention in the responses, though, this is not fully satisfactory criticism, as I acknowledge that much of the related literature apparently uses only ER graphs in their experiments.

**Limitations:**

yes

**Quality:**

3

**Strengths And Weaknesses:**

**Strengths:**

- Novel identifiability results for hypergraphical models.
- Novel algorithm for learning hyper DAGs.
- Empirical evidence that their algorithm indeed work better than others in a 2D setting for additive models.
- In general, the paper is well-written and discusses related work appropriately.
- Helpful illustrations.

**Weaknesses:**

Experiments are only performed on sparse Erdős–Rényi graphs over 30 vertices, making them somewhat limited in my opinion. As the authors argue in their abstract about "abundance of real-world processes exhibiting higher-order mechanisms", I would have hoped for more comprehensive experiments or some other kind of evidence about the significance of the work.

Some terminology is not explained and may take some effort even for a knowledgeable reader to undestand how they generalize to the hypergraphical setting, e.g., immoralized hypergraph of an HDAG. Sometimes, the explanations also come relatively late, e.g., multicolliders are introduced only 13 lines after Theorem 1 even though they are already needed for the theorem statement.

**Minor:**
- Missing punctuation after numbered equations
- Line 76: now unordered edges are ordered pairs $(i, j)$. Should they instead be sets $\lbrace i, j \rbrace$?
- Def. 1, 3, 5: I'd add some literature reference here.
- Eq. (6), (7): inconsistent use of italics in $Pa$.
- Eq. (8): should $Pa$ be $HypPa$?
- Line 148, 261: missing parentheses for citations.
- Line 163: missing whitespace.
- Line 299: add citation to "Following previous works".
- Line 576: broken reference.

---

> ### Author Rebuttal · Authors · 2025-07-31
>
> Thank you for your review of our work, we briefly highlight the key strengths listed before addressing the weaknesses.
>
> ## Strengths
>
> > Novel identifiability results for hypergraphical models.
>
> > Novel algorithm for learning hyper DAGs.
>
> ## Weaknesses
>
> > Some terminology is not explained and may take some effort even for a knowledgeable reader to understand how they generalize to the hypergraphical setting.
>
>
> We apologize that there is a lot of terminology which felt difficult to understand.  We chose to introduce the definitions of body and multicollider during Theorem 1 because of their extremely close correspondence with skeleton and collider.  We had hoped this would give easier intuition for readers who were already knowledgeable about these existing definitions.  Based on the feedback from multiple reviewers, we understand this was not the case and we have added to Table 1 a hyperlink to a new section of the Appendix which more formally states these definitions.
>
> To reiterate, the **body** of the HDAG takes the directed hypergraph and replaces every directed hyperedge with its undirected version.  This is written as $H’ = \\{ S \cup \\{ j \\} : (S,j) \in H\\}$.  A **multicollider** of an HDAG is any set of $n$ parents and one child.  A multicollider is unshielded if the body of the hypergraph does not also contain the hyperedge with all $n$ parents.  The intuition for this is that the only time all $n$ parents interact with one another is in creation of the normalizing constant $Z$ in equation 8.  This is a special interaction which disappears under marginalization of the child, explaining why we cannot condition on colliders or descendants of colliders in classical DAG theory.  (Please also note our correction to your question about Equation 8 below.)  The **immoralized hypergraph** is the undirected hypergraph containing the body of the HDAG along with all of the $Z$ terms from Equation 8, i.e. the hyperedges created by simultaneously marrying all of the parents.  It is hoped that how these terms correspond with existing terms is now more clear and we would be very happy to explain any further confusions regarding this.
>
> > As the authors argue in their abstract about "abundance of real-world processes exhibiting higher-order mechanisms", I would have hoped for more comprehensive experiments or some other kind of evidence about the significance of the work.
>
> It is unfortunate that you felt misled about the nature of the experiments.  This sentence is taken by the authors to mean that, despite the abundance of works showing that real-world processes do exhibit higher-order mechanisms [A, B, C], there remains to be sufficient treatment of **how** to learn such mechanisms in a causal framework.  In our results, we find that the situation is even more dire than one might expect.  Even in the most common setting (ER graphs), with the simplest adjustment to include higher-order mechanisms, most existing algorithms not only fail to learn the higher-order causal structure (Table 4), but also fail to learn the typical causal structure in the presence of these higher-order mechanisms (Tables 2 and 3).
>
> These negative results point to the idea that maybe the reason causal structure learning cannot be applied to real-world systems is because existing algorithms are not able to handle higher-order phenomena.  Our theoretical results then confirm that higher-order structure can be learned directly from the data, implying that we need to do work on improving the algorithms.  Our proposed method HCAM is a first attempt in this direction.  Our work can be seen as a similar result to [D], which provides negative results to emphasize that current synthetic data doesn’t represent fully realistic settings.
>
> In this sense, even if one only tentatively believes in the abundance of higher-order mechanisms, what we feel has been clearly demonstrated by the current experiments is that existing methods will not be able to capture them in the situations where they do arise.
>
> > Experiments are only performed on sparse Erdős–Rényi graphs over 30 vertices, making them somewhat limited in my opinion.
>
> Four of the six previous works we compare against also focus on ER graphs with a relatively small number of vertices.  We agree that this comes with limitations, but it is also the fairest way to compare against existing algorithmic proposals.
>
> Our larger concern is that even with very simple multilinear terms, existing algorithms fail on this limited setting.  Additionally, we feel the limitation to 10K samples may also prove to be a key reason why algorithms fail in this setting.  We are currently running experiments in this setting for HCAM (other algorithms do not scale well to this regime, which is why we did not originally compare in this setting in the first place).
>
>
> [A] “Networks beyond pairwise interactions: Structure and dynamics”. Federico Battiston et al. 2020.
>
> [B] “Higher-order genetic interactions and their contribution to complex traits”. Matthew Taylor et al. 2014.
>
> [C] “Dynamics on higher-order networks: a review”. Soumen Majhi et al. 2022.
>
> [D] “Beware of the Simulated DAG! Causal Discovery Benchmarks May Be Easy To Game”. Alexander G. Reisach et al. 2021.
>
>
> ## Minor Points
>
> > Def. 1, 3, 5: I'd add some literature reference here.
>
> We have added references to “Probabilistic Graphical Models: Principles and Techniques” (defn 4.11/4.12 and defn 3.4) for the first two definitions and “Causal Discovery with Continuous Additive Noise Models” (equation 6) for the third definition.
>
> > Eq. (6), (7): inconsistent use of italics in Pa.
>
>
> Italics removed from Eq (6).
>
> > Eq. (8): should Pa be HypPa?
>
>
> No, it should not.  Although the functional structure of how the child is generated according to different sets of hyperparents will vary with the chosen structure, the normalizing constant will in general depend on all possible parents in order to form a valid conditional probability.
>
> > Line 299: add citation to "Following previous works".
>
> We have added a citation to CAM, RESIT, SCORE, and BOSS which all use ER graphs.

---

> > ### Comment · Reviewer_nSx3 · 2025-08-01
> >
> > Thank you for the detailed rebuttal!
> >
> > While I acknowledge that Erdős–Rényi graphs are prominent in the related literature, I'm still critical of them being the only instances as randomized structures can easily lack structure that real-world systems have and that other methods might be able to exploit. Thus, I feel like to argue about real-world significance of the method, the experiments should utilize at least some real-world instances and not just randomized ones.

---

> > > ### Author Response · Authors · 2025-08-01
> > >
> > > Thank you for your honest response.  If you could provide some references showing how to use real-world instances for causal discovery, it would be very helpful for us.

---

> > > > ### Comment · Reviewer_nSx3 · 2025-08-04
> > > >
> > > > Thank you for the response. I am less familiar with causal learning benchmarks, especially for additive models, for which you need fewer assumptions than in general for identifiability (i.e., generally purely observational data is insufficient).
> > > >
> > > > I acknowledge that this may be a bit disappointing response, since I am essentially asking for something for which I cannot provide a good reference, but I nevertheless feel like I cannot increase the significance score without some sort of real-world experiment. The AC may, of course, overrule may subjective opinions if they wish so.

---

> > > > > ### Author Response · Authors · 2025-08-04
> > > > >
> > > > > Thank you for your continued honesty.  We have the feeling that real-world experiments are much less common for causal discovery methods because there is no way to evaluate any of the typical metrics.  Either way, we can give thought to how to present a better experiments section.
> > > > >
> > > > > (p.s. Not very important but just to be certain the key reason for identifiability is the continuous variables with the additive *noise* modeling (ANM) assumption. With or without the additive models we focus on does not really change the assumptions.)

---

### Official Review · Reviewer_YZBu · 2025-06-24

**Clarity:** 3
**Significance:** 2
**Originality:** 2
**Rating:** 3
**Confidence:** 4

**Summary:**

The submitted work addresses causal structure learning with directed acyclic hypergraphs under causal sufficiency. The author generalizes the notion of the Markov property of simple (un)directed graphical models to hypergraphs. They prove that for an ANM the corresponding hypergraph is also identifiable and extends the idea of a Markov equivalence class (MEC) to hyper MEC.
For Gaussian ANM they propose a greedy algorithm (based on CAM and SIAN) for learning  the hypergraph  that corresponds to the SEM.

**Questions:**

Thank you for your interesting work addressing interaction effects in causal discovery with hyperedges. During the review of your submitted paper I collected the following questions, remarks and typos and enumerate them to facilitate addressing them in the rebuttal.


**Questions (Q)**
1. What are the formal definitions of body and multi-colliders for the hypergraphs? (only indirect definition in l. 186 and l. 191f)
2. What is the difference between a DAH and a DAG that introduces a latent variable for each hypergraph (the terms of body and multicollider would be not needed then)?
3. How did you derive the number of 432 HDAGs that belonging to the MEC of a fully-connected DAG with four nodes? Did you allow for the possibility that a hyper-edge is present between three nodes, but a standard edge between two nodes is missing?
4. What are the details for the data generation? In particular:
5. How do you sample hyper-edges? Do you generate multigraphs (i.e. allowing for both a hyperedge and a standard edge)?
6. What are the precise distribution for sampling the coefficients and variances of the Gaussian?
7. How is the higher-order SHD computed?
8. What is an immortalized hypergraph?

**Remarks (R)**
1. I would consider the formulations in Eq. 4 and 6 to be more general than E. 5 and 7, since the later contain more information (more independent factors instead of a single one containing all neighbors/parents)
2. Figure 1 is not very informative and at the same time slightly confusing, an ANM does not specify interactions are represented in a graphical model,
the key information consists in the visualization of interaction, instead of showing the first row, the  true SEM should be displayed (-> $X_4 = f_{12}(X_1,X_2) + f_{23}(X_2, X_3)$ and $X_6 = f_4(X_4) + f_5(X_5)), the column (for the remaining second row) should list the different graphical models  (DAG, DAG with multiple hyperedges for the parents, DAG with a single  hyperedge for the parents)
3. The term hyper-Markov equivalence class slightly abuses the definition of the Markov Property, introducing a latent variable for each hyperedge may elegantly resolve the issue
4. Directed acyclic hypergraphs are abbreviated in other works by DAH
5. The note on unprimed graph notation in parentheses can be omitted (l.78) saving space for other content.
6. It may be more intuitive to explain how a hyperedges arises from a SEM.
7. Eq. 2 is recursively defined for the nodes (stated from the perspective of node i).
8. Figure 2 takes quite some time to understand, this is partially due to the second block that already contains the distinction by ‘body’, I would appreciate a third body in the second block (each placed below each other) to help the reader understand the idea (smaller undirected hypergraphs to meet the space constraints should work fine), visible nodes would also help
9. The number of runs for the experiments and clarification of the stated uncertainty (std) is only provided in the checklist.
10. The zero basline is clearly favored by the sparsity of the DGM.
11. The presentation of the work would benefit from an overview of the algorithm (i.e. pseudocode).
12. Highlighting in tables by using a bold font style is inconsistent, the following seems for reasonable:
Table 2: 1D: *CAM*, 2D: *HCAM*, 3D: *BOSS, GES, zero, HCAM*
Table 3: 1D: *CAM*, 2D: *BOSS, HCAM*, 3D: *BOSS*
Table 4: 1D: *CAM*, 2D: *HCAM*, 3D: *BOSS*
13. The good performance of BOSS in Table 2 and 3 is not discussed


**Typos**
 - Eq. 6: \mathrm{Pa}
 - l.170 “rather than the sum of *more 1D or* 2D energy terms”
 - Figure 2: in the third block it should read HDAGs (or DAH) instead of DAGs
 - l.211: Theorem 27 (28 is a proposition)
 - l.576: missing reference for the Eq.

**Ethical Concerns:**

["NO or VERY MINOR ethics concerns only"]

**Final Justification:**

**I cannot recommend an acceptance of this submission.**

To account for the rebuttal of the authors in which they sucessfully addressed the questions and some of the listed weakness, I raised my overall score to weak reject, but increased the confidence of my evaluation. The promised changes are suitable to increase clarity and reproducibility of this work, but are quite amble (with the consequence that for some only a high level description was provided and a leap of faith is required for the changes).

Overall the submitted work investigate an interesting idea, which is arguably straight-forward to the research community in this field. It remains very limited by the requirement of ANMs and the performance of the proposed algorithm. The concern of reviewer *iobK* regarding applicability and scalability are shared, while the criticized lack of real-world experiments by reviewer *nSx3* is not, although it would certainly strengthen the presentation and work.\
In addition, the experimental evidence is rather weak. The reviewer disagrees with the author that the statistical analysis is valid (3 runs representing a full distribution do not support the statements on statistical significance). This issue was only revealed by the reported decimals after a very close inspection and a final confirmation at the end of the lengthy discussion with the authors, who neither communicated it in their submission nor the initial rebuttal, but have to be well aware of their evaluation.

| Criterium | Change |
| --- | --- |
| Quality | $1 \uparrow 2$ |
| Clarity | $2 \uparrow 3$ |
| **Rating** | $2 \uparrow 3$ |
| **Confidence** | $3 \uparrow 4$ |

**Limitations:**

1. The assumptions for causal discovery are openly addressed in subsection on undirected Models [l.79ff], however they apply also to directed graphs and, hence could be already introduced at the beginning of the whole section preceding the two subsections.
2. The simplicity of the multi-linear model with additive Gaussian noise for 2D and 3D is mentioned [l.326], but the limitation of the provided experimental evidence is not addressed.
3. The problem of sparsity is mentioned indirectly by the *zero* baseline, but not addressed by experiments on denser graphs.

**Paper Formatting Concerns:**

- Vertical lines in Table 1
  - Checklist: For the current version, the answers to questions 4, 5 and 6 in the checklist are simply incorrect. Experiment data or code was clearly not provided for the review, neither is the documentation of the experimental section complete.

**Quality:**

2

**Strengths And Weaknesses:**

**Strengths (S)**
 1. Theoretical proof of the identifiability of the causal hypergraph corresponding to an ANM,
 2. Reduction of the number of Markov equivalent graphs to the number of hyper-Markov equivalent hypergraphs.


**Weaknesses (W)**
 1. Related literature on directed acyclic hypergraphs is not discussed, e.g. [1,2,3],
 2. Very limited experimental evidence (only sparse ER graphs),
 3. Insufficiently documented data generation process and used hyperparameters for the comparison of algorithms (can be easily addressed in  the appendix, but as presented now the experiments are not reproducible and resulting in a lower overall score),
 4. Results for the higher-order SHD and SID are very bad for all models, even the proposed algorithm (limiting its applicability even in the multilinear Gaussian case),
 5. Clarity of the presentation (can be easily addressed),
 6. The proposed algorithm does not test for independence, the reasoning with a refinement of the Markov equivalence classes does not hold,
 7. For the stated theorems no proof nor sketch of it is provided in the paper, only in the Appendix
 8. No simple experiments investigating why the learning of higher interaction effects does not work,
 9. No ablation study for the importance of key components of the proposed algorithm,
10. Listed limitations below.


[1] Javidian, Mohammad Ali, et al. "On a hypergraph probabilistic graphical model." Annals of Mathematics and Artificial Intelligence 88 (2020)

[2] Ma, Jing, et al. "Learning Causal Effects on Hypergraphs." 28th ACM SIGKDD Conference on Knowledge Discovery and Data Mining, KDD 2022.

[3] Evans, Robin J. "Graphs for margins of Bayesian networks." Scandinavian Journal of Statistics 43.3 (2016)

---

> ### Author Rebuttal · Authors · 2025-07-31
>
> # 1.
>
> Thank you for your detailed critique of our work. We will address each of your many points in what we believe to be descending order of importance. Reiterated questions are later removed due to space.
>
> > W1: Related literature on directed acyclic hypergraphs is not discussed, e.g. [1,2,3],
>
> Thank you for these references with which you are familiar, we will add them to our work. We note that [2] is a pretty unrelated framework, creating a hypergraph over individual subjects for ITE rather than over variables.  [3] is more related, but in their Figure 1, it is clear to see that they focus entirely on adding hypergraph structure to the latent marginalization. Under the causally sufficient setting we study, their work would reduce to a DAG. An adequate extension to include marginalizing and conditioning over latent variables for HDAGs would likely encompass both works (discussed further in the next paragraphs).
>
> The last work, [1], is much more nuanced in its differences. In this work, they are specifically focused on merging the directed and undirected graphs under the LWF interpretation of the chain graph. This is different from our framework which focuses entirely on extending the DAG framework. We will call ours HDAG and theirs DAH throughout our response to avoid further confusion.
>
> Using their language, we can say the “shadow” of a DAH is always a chain graph whereas the shadow of an HDAG is always a DAG. This is exactly the correspondence depicted in our Figure 1 where the shadow is the same for 1a, 1b, and 1c.  If our understanding is correct, their Equation (7) on page 12 is an extension of our Definition 4. Their work does not handle continuous variables or additive noise models. Despite this generalization to chain-graph-like structures, they do not provide the appropriate generalizations of Markov characterization or identifiability results of the DAH. In particular, they only show each DAH respects the Markov properties of its LWF chain graph shadow. It is imagined the identifiability results herein could be generalized to also handle the DAH case. Nonetheless, given that the three major interpretations of chain graphs (including marginalization) have remained separated for decades, it seems unlikely this would overlap with the interpretation from the above paragraph. Lastly, it is noted that no practical experiments or algorithms are available in relation to the DAH.
>
> > R4: Directed acyclic hypergraphs are abbreviated in other works by DAH
>
> This turns out to be quite convenient, because as discussed they use a different definition than ours. As mentioned above, we will continue to use the two different abbreviations DAH and HDAG to make it clear which work we are talking about throughout our reply.
>
> > Q2: What is the difference between a ~~DAH~~ HDAG and a DAG that introduces a latent variable for each hypergraph?
>
> > R3: The term hyper-Markov equivalence class slightly abuses the definition of the Markov Property, introducing a latent variable for each hyperedge may elegantly resolve the issue
>
> Unfortunately, no, the naive addition of latent variables is not sufficient for our setting. Imagine adding a LV for each hyperedge. At the node of the LV, you could have a variable representing the correct functional term; however, you would need to have a special node to make sure that no additive noise occurs. Next, you would need to aggregate all of the functional terms into the next observed node and use additive noise. However, using the typical ANM structure would destroy the additive structure when combining the LVs; therefore, we would need to use a special edge implying CAM-style aggregation. Instead of adding a special node type and a special edge type, it is much more natural to introduce the hypergraph structure. The situation is similar for the classical setting.
>
> Accordingly, we hope that it is now clear how our studied regime differentiates itself from the existing works using hypergraph structure and how our setup and theoretical results are a unique contribution to the causality literature compared with those related works. Moreover, our setting is not trivially the same as other well-known settings like considering the marginalizing and conditioning of latent variables. If there are any remaining points of confusion here, please let us know immediately so that we may attempt to address them. Otherwise, we will continue to the next section to discuss the listed strengths and technical misunderstandings of our work.
>
> # 2.
>
> ## Strengths
>
> > The submitted work addresses causal structure learning with directed acyclic hypergraphs under causal sufficiency
>
> > Theoretical proof of the identifiability of the causal hypergraph corresponding to an ANM
>
> In addition to this, we have the identifiability result in the classical setting as well, giving identifiability at least up to the MEC and in some cases even more than this.
>
> ## Technical Points
>
> > Q1, Q8
>
> We have added another section to the appendix extending Table 1 to make sure the definitions of all the introduced terms are clear. See response to reviewer "nSx3".
>
> > R1
>
> You are correct about this, but unfortunately much in the same sense that writing log p(x) = \xi(x) for some function \xi(x) is even more general than all four of these equations. In this case, we would just be doing probabilistic modeling instead of causal structure learning.
>
> It is understood that you already know the usefulness of causal structure for understanding intervention effects. There are many reasons to also learn the mechanism structure, not least of which is that they are known to exist in a variety of physical phenomena [A, B, C]. Contrast this with our work's experimental results showing that existing algorithms cannot learn these mechanisms (even failing to learn the causal structure in this case!). If this more specific structure is identifiable directly from the data, then why shouldn’t we incorporate it? If this structure is also learnable, then why can’t current algorithms succeed in these simple cases?
>
> > L1
>
> Indeed, as stated we make the assumption of causal sufficiency throughout the work, applying to all three subsections. Extending beyond this setting would likely be a great undertaking.
>
> > W6
>
> We believe that this must be a misunderstanding of some kind. The ideas of Markov equivalence are from our second setting like in Section 2.2, whereas the proposed algorithm is applied to the third setting from Section 2.3.
>
> Lastly, we will get into the minutiae of the concerns regarding the experimental setup and the writing.
>
> # 3.
>
> ## Experiment and Dataset Concerns
>
> > W2, R10, L3, W4, L2
>
> We group these concerns because they are all focused on the same topic. We first emphasize that sparse ER graphs are an extremely common setting, employed by 4 of the 6 baseline methods which we compare against. Focusing on this setting does not significantly limit the findings or invalidate the use of the baseline.
>
> It is also reminded that we provide experimental evidence in terms of a negative result, demonstrating that most existing algorithms cannot handle 2D or 3D data even for simple ANMs with Gaussian noise. Due to this, the simplicity of the multi-linear model is actually an advantage to the empirical evidence, rather than a limitation.
>
> > W3, Q4, R9
>
> Data generation details are in W2 of “35ca”. Higher-order SHD is computed as the SHD of the bodies, where every algorithm except CAM is assumed to include all interactions like in Figure 1f to enable a comparison.
>
> > W8
>
> Although we point to several works supporting our stated hypothesis that the largest reason is the small sample size relative to the exponentially large modeling space of probability densities, we agree that an experiment specifically verifying this would enhance the empirical contribution. A key challenge is that existing methods do not scale to 100K+ samples. We are running experiments now (at least for HCAM) on these larger datasets.
>
> > W9
>
> This is another point which could enhance the empirical understanding of the HCAM algorithm. Is it suggested that we ablate parts of the algorithm with oracle knowledge?  Since no other existing algorithms can handle HO structure, it is not straightforward to choose which aspects of the algorithm can be ablated. This is especially so because most of the work in the CAM and HCAM algorithms are done in step 2.
>
> ## Writing and Presentation Concerns
>
> > W5
>
> We hope with the adjustments to the writing made in accordance with the technical misunderstandings discussed in the previous part of our response, as well as the other points below, the clarity has been greatly enhanced.
>
> > R2
>
> We apologize for the confusion, we have added the nonparametric SEMs to the bottom three panels of Figure 1. We agree that this will help to make Figure 1 more easily understandable.
>
> The ANM does not explicitly specify the interactions in a graphical model. The point we are making here is that it does so implicitly. As mentioned above, the CAM or ANM assumptions lift to different HDAGs (a to d or c to f; our generalized version in the middle), but the shadow of all three HDAGs is the same DAG.
>
> > R8
>
> We have added visible nodes and therefore colored in the ellipses. This visually helps a significant amount. We have also added a third picture to the second block.
>
> > R11
>
> We have added algorithm pseudocode to the appendix section with the algorithm details.
>
> > R12, R13
>
> We have bolded according to the logic “at least as good as the baseline which assumes independence”. We have added a sentence to make this clear and adjusted the current bolding slightly. BOSS’s good performance on 3D is not statistically significant, but we have added a sentence addressing this.
>
> > Typos
>
> Everything resolved except for line 170 which is written as intended. We have added the equation $\theta_{ij}(x_i,x_j) + \theta_{ik}(x_i,x_k) + \theta_{jk}(x_j,x_k)$ to make the meaning more obvious.

---

> > ### Comment · Reviewer_YZBu · 2025-08-04
> >
> > *I appreciate the rebuttal of the authors, the promised changes will enhance clarity of the presentation of their work.*
> >
> > Remark **R9** (number of runs) was not included in the mentioned response to the other reviewer. Only the checklist of the submission reveals that 3 independent runs were conducted. Further inspection of the decimal values in Table 2,3 and 4 suggest that only a single graph and parameter coefficient (=1 data set) for each run was considered.
> > If this is correct:
> > Which 3 graphs with 30 nodes were sampled? Please state the adjacency matrices in the appendix along with the sampled coefficients.

---

> > ### Comment · Reviewer_YZBu · 2025-08-04
> >
> > While I disagree with the stated descending order of importance, for the rest of my response I follow the authors’ ordering into three blocks for clarity:
> >
> > **(1)**\
> > **W1:** I did not claimed that there were no fundamental differences between [1,2,3] and the submitted work. Nevertheless, they are related by the use of hyperedges for probabilistic graphical models and, hence, should be at least discussed as *related* work.
> >
> > **Q2:** I agree with the authors that a different node type for the proposed (auxiliary) random variables would have to be added, since they result from deterministic functions without additional exogeneous random noise.
> > Nevertheless, they are random variables for which independence relations could be investigated as suggested in **R3**. In fact, in the beginning of section 2.2., the authors already treat the hyperedge as a latent representation with one “ our arrow” and several “in arrows”.
> >
> > **(2)**\
> > **Q1, Q8:**
> > While reviewers and some readers who are also very familiar with the topic, may directly understand Table 1 that lists analogies between DAG and HDAG terms, it is good practice to formally introduce new definitions avoiding room for any ambiguity.
> > Adding the definitions in the appendix, will provide some clarity, however, I strongly recommend a brief definition in section 2.2 of the main text (also given the extra page).
> >
> > **R1:** I think the authors completely missed the point of this (minor) remark. I consider the model with hyperedges to be more fine-grained, since it naturally encodes more information about the causal structure. Only in this sense, it is more general.
> > I disregard the second paragraph
> >
> > **L1:** As stated and agreed by the authors, causal sufficiency is a assumption that applies to both, undirected as well as directed graphs in their work. For clarity, it
> > > could be already introduced at the beginning of the whole section preceding the two subsections.
> >
> > **W6:** For clarification, the authors did not claim that their algorithm refines the Markov equivalence class by independence tests.

---

> ### Comment · Reviewer_YZBu · 2025-08-04
>
> **(3)**\
> **W2, R10:**
> To start with, 4 out of 6 does not make an “extremely common setting” (no proper reasoning).
> > Focusing on this setting does not significantly limit the findings or invalidate the use of the baseline.
>
> Without doubts, the experimental evidence is limited to the test setting, namely sparse ER graphs. The advantage of sparsity favoring methods (e.g. zero baseline) is an artefact of this setting.
>
> **L2, W4:**
> For the 2D setting, experimental evidence that the proposed method HCAM is competitive with some other baselines is only provided for the multilinear DGM with additive Gaussian noise, hence, it is certainly a limitation of the positive result.
> For the 3D setting, the authors transparently report a negative result of HCAM, since it performs exactly as the zero baseline. Although the other algorithms do not do better, it remains (also) a limitation for HCAM. The fact that both GES as well as HCAM reduces to the zero baseline should be discussed by the authors (only implicitly hinted by the very same numbers).
> In conclusion, both limitations do exist.
>
> **W3, Q4, Q5, R9:**
> The generation of the graphs needs some further details to avoid ambiguity:
> - For nodes of a sampled ER-graph with more than two or three variables: Are there exclusively multiple hyperedges sampled (overlapping hyperparental sets)?
> Let’s assume $X_1, X_2, X_3$ (random order) are the parents of some node in a sampled ER graph and the 2D setting is considered. Are the set of hyper-parents for the respective hyper-edges then $(X_1, X_2)$ and $(X_2, X_3)$ as in Figure 1e)? Is this meant by a cyclic fashion for a given random ordering?
> - For the 3D model, are there any additional 2D hyperedges (when only two parents are present)? Conventional edges only exist when only a single parent is present?
> For sampling of the parameter, please clarify the (non-standard) generation:
> - Why do you sample from the log uniform distribution and not the (standard) uniform distribution?
> - Why do you rescale the coefficients by the square root of both expected moments?
> - Why do you additionally divide by the total number of parents?
>
>
> **Q7:**
> As stated, the higher-order SHD is computed on the hypergraph bodies (hyper-edges as well as undirected standard edges).I noted that the result of both SHD are exactly the same for CAM in the 1D setting. Does this imply that the standard SHD computes only the distances between skeletons?
> In addition, why is for 1D the higher-order SHD for HCAM smaller than the standard SHD?
>
> **W8:**
> A very simple setting with a single graph of four nodes and a single hyperedge (three parents and their child) and parameters values close to the mean of the their distribution may be provide some insights (although clear simplification).
>
> **R12:**
> The stated logic is not accurate, since the baseline itself is sometimes also bolded. In addition, the uncertainty over multiple runs (standard error of the mean) should be considered as well. Here, the validity of std over 3 runs have to be discussed.
>
> **R13:**
> > BOSS’s good performance on 3D is not statistically significant, but we have added a sentence addressing this.
>
> Firstly, an added sentence should be stated in the rebuttal as a direct quote.
> Secondly, the author’s notion of “statistical significance” needs some further explanation, in particular in the case that only three sampled graphs and a single set of parameters were tested.
>
>
> *General note:*\
> Due to the changed rebuttal policy that does not allow to update the pdf, I would welcome a summary of added paragraphs or sections (both main paper and appendix) as a response to all reviewers. The added results are already sufficiently documented.
>
> *Polite reminder:*\
> From the review there is a question (*Q3*) as well as a weaknesses (*W7*) that were not addressed by the authors in their rebuttal, but they may still want to do so. By contrast, *Q6* has been addressed, although not clearly marked.

---

> > ### Author Response · Authors · 2025-08-07
> >
> > ### (0)
> >
> >
> > Thank you for your continued dedication to improving the work.  We agree that many of the writing and presentation changes you suggested will help clarify the communication of the proposed setting.
> >
> >
> > For **R9**, we indeed use three different datasets for each run.  Although adding all the 30x30 adjacency plots to the appendix seems a bit extreme, we can include the .json files with both the adjacencies and the multilinear coefficients to the code release, and because random seeds were set they should also be regeneratable.
> >
> >
> > ### (1)
> >
> > **W1**
> >
> >
> > We partially interpreted the first weakness to mean the primary weakness.  We were not previously aware of these works and given the originally assumed expertise of the reviewer, this required considerable effort to drill into the details for the differences with [3].  We hope the differences and the significant contribution our work makes to [2, 3] remains clear.
> >
> >
> > **Q2**
> >
> >
> > This is seemingly the last major part about the writing where we actually disagree with the reviewer.  We do not believe that there is a reasonable correspondence with the latent variable interpretation, given the need to add both special nodes and special edges.  Moreover, even if willing to add these special requirements, looking only at independence relations would again reduce to existing theory, the same simplification made by [3].  The key innovation for the new identifiability result in this work was in using multi-independence relations.
> >
> >
> > This confusion is especially concerning because we imagine that future work should be able to extend beyond the case of observed variables, as discussed in our original response to W1.  Would a sentence warning against this presumption about LVs be sufficient (e.g. adding this note to line 105)?
> >
> >
> > ### (2)
> >
> >
> > **Q1, Q8**
> >
> >
> > If granted an extra page, we would certainly include the definitions in the main text as well.  Not adding formal definitions to the appendix initially was indeed an oversight.
> >
> >
> > **R1**
> >
> >
> > It seems we misinterpreted the intention behind this remark.  We are in agreement with you that our work is more general than previous approaches.
> >
> >
> > **L1**
> >
> >
> > Yes, the original L1 seems to group both a limitation of the work and a remark about the writing.  To clarify, causal sufficiency is assumed throughout all three sections (not two), and positivity is only required for simplicity, mostly for settings 1 and 2.  Nonetheless, we had already moved this paragraph one paragraph higher and were debating if further rewriting for clarity is required.
> >
> >
> > **W6**
> >
> >
> > We provide no algorithm for setting 1 or setting 2.  These were used as theoretical stepping stones for our setting of interest.  We only provide an algorithm for setting 3 where the DAG and HDAG are identifiable exactly.  As a reminder, our Theorem 4 shows the equivalence of settings 2 and 3 under Gaussian errors.

---

> > > ### Author Response · Authors · 2025-08-07
> > >
> > > ### (3a)
> > >
> > > **W2, R10**: To start with, 4 out of 6 does not make an “extremely common setting” (no proper reasoning)...
> > >
> > >
> > > Although we agree that, in isolation, 4 out of 6 might not mean “extremely common”, we note that each of the 2 out of 6 are over twenty years old.  You may be disappointed to hear that other common settings include ER1, an even sparser graph than studied, as well as the scale-free graph, another sparse random graph.  Although we feel strongly that the chosen test setting does not greatly limit the findings, we can understand how the quantity of experiments could raise doubts, see further discussion below.
> > >
> > >
> > > On the baseline, we were never in disagreement that assuming no edges is implicitly favored by sparser datasets.  Perhaps the missing piece of information is that these are already the standard settings for verifying causal discovery.  If you instead prefer, you can imagine we do not include this baseline comparison.  Our method is then always state-of-the-art for the new settings we introduce.  We did not feel this would present a complete picture of the behavior.
> > >
> > >
> > > **L2, W4**: For the 2D setting, experimental evidence that the proposed method…
> > >
> > >
> > > Again, the focus on the success and failing of HCAM is different from our intended focus.  Our major intended focus is about the lackluster performance of all algorithms.  The secondary point is about the success of HCAM on 1D and 2D.  We perhaps take for granted that causal discovery focuses on sparse graphs (see baseline discussion above).  With this understanding of the literature, we hope it is more clear how the simpler setting (multilinear + Gaussian) is actually advantageous for the claims we are making.  Although we agree that seeing the same results on alternate DGP choices would strengthen all of these claims, especially the positive claims about success, we don’t think the simplicity or sparsity can themselves be seen as weaknesses.  To clarify, we understand that ‘both’ limitations refers to: (a) the sparsity of the ER4 DGP favoring the sparse baseline; and (b) the use of a single DGP limiting the generality of empirical claims.
> > >
> > >
> > > **W3, Q4, Q5, R9**: The generation of the graphs needs some further details to avoid ambiguity
> > >
> > > That is exactly correct, we would have $(X_1, X_2)$ and $(X_2, X_3)$ like in Figure 1e.  To make it absolutely explicit, in the 3D case for ordering $X_1, X_2, X_3, X_4, X_5$ we would have $(X_1, X_2, X_3)$, $(X_2, X_3, X_4)$, and $(X_3, X_4, X_5)$.  Any subset of a hyperparent always “exists” because e.g. a 2D function can always represent any 1D functions, see proof of Thm 3.  However, to your more likely question of did we add terms specifically accounting for these hyperedges, the answer is no.  Moreover, in cases like a 3D model with only two parents, we actually chose to remove the adjacencies entirely, as to not dilute the ‘dimensionality’ of the dataset.
> > >
> > > On our “non-standard” choices for the DGP: The log-uniform is to sample around $+/-$ 1.0 to keep the coefficients bounded away from zero.  The rescaling is necessary to avoid later variables having significantly higher variance than early variables, which would lead to a causal dataset which is too easy to solve because the causal order is mostly dictated by the variances.  This has been called “varsortability” by previous work [4].  Assuming $X_1$, $X_2$, and $X_3$ are three Gaussian variables with unit variance, the term $X_1 * X_2 * X_3$ has variance 1.0 if they are perfectly independent and variance 15.0 if they are perfectly correlated.  Early on in the DAG generation they will be the former and later on in the DAG generation they will be the latter.  Renormalization by number of terms is done for the same reasons.  Ultimately, these are empirical choices made to balance the variance of the DGP.  For the small D=30, these choices might be on the heavier side of rescaling to ensure the data is not too easy.
> > >
> > >
> > > [4] “Beware of the Simulated DAG! Causal Discovery Benchmarks May Be Easy To Game”. Alexander G. Reisach et al. 2021.

---

> > > > ### Author Response · Authors · 2025-08-07
> > > >
> > > > ### (3b)
> > > >
> > > > **Q7**
> > > >
> > > > Yes, this astute point about the SHD and higher-order SHD is confusing at first glance.  It is first reminded that SHD, an existing metric, is computed on the directed graph not on the undirected skeleton.  The higher-order SHD is computed on the undirected body (which you are right to identify is the skeleton in the 1D case).  The HO SHD is defined this way to focus on capturing the hypergraph structure, rather than overlapping heavily with the existing SHD metric.  From the SID=0.0 of CAM, we can interpret that it never misorients an edge.  This is not the case for HCAM and the drop in HOSHD for HCAM reflects the edges in the skeleton which it had misoriented.
> > > >
> > > >
> > > > **W8**: A very simple setting with a single graph of four nodes and a single hyperedge (three parents and their child) and parameters values close to the mean of the their distribution may be provide some insights (although clear simplification).
> > > >
> > > >
> > > > Yes, this simplified setting allows for clearer insights into the performance of the algorithm.  We briefly note that the originally mentioned experiments on 100K samples did not improve the performance.  We believe this means even 100K is not enough samples for thirty nodes.  This claim will be further supported after the next paragraph discussing the results of the simplified setting with four nodes.
> > > >
> > > >
> > > > We run the suggested experiment.  We generate 30 seeds where we force the hypergraph to have the suggested three parent structure (still multilinear).  We then vary the number of training samples provided from N=10 to N=100,000.  We plot both the success rate of recovering the body (HO SHD =0.0)  and of full recovery (SHD, SID, and SO SHD all zero) as a function of N.  The success rates increase as we increase the number of samples.  After 1000 samples, the algorithm already recovers the body with >90% success and at 10,000 the success rate is 100%.  For the recovery of the directed graph, however, the success rate grows, but even at 100,000 samples only grows to a 63% recovery rate.  This provides strong empirical evidence that the claimed reasoning of the paper indeed is the true reasoning for algorithm failure.
> > > >
> > > >
> > > > **R12**
> > > >
> > > >
> > > > Explicitly following the stated logic, we would actually need to bold the baseline every time.  We have instead chosen to remove the bold from baseline when other methods clearly outcompete the simple baseline.  The erroneous bold from this logic is only possibly the baseline from 2D in Table 2, which we have updated.
> > > >
> > > >
> > > > **R13**
> > > >
> > > >
> > > > The meaning of three runs is discussed above.  Error bars are 1 sigma.  See suggested changelist separately.
> > > >
> > > > **General note**:
> > > >
> > > >
> > > > In the near future, we will provide a changelist of the writing to allow for an easier updated evaluation, even without an updated PDF.
> > > >
> > > >
> > > > **W7**
> > > >
> > > > Is it preferred that we move line 249 to the beginning of the section? Theorem 1 is already followed by two paragraphs of proof sketch and intuition.  Theorem 2 is a known result.  We can add a sentence to Theorem 3 that it uses a similar proof technique to Theorem 2, but Theorem 4 also provides an intuitive description of the proof strategy.
> > > >
> > > >
> > > > **Q3**
> > > >
> > > > For the DAGs in the far left panel, it is fairly clear to see there is a correspondence with the 24 orderings on 4 elements.  Thus, we can reduce 432 to 18 by this symmetry.  Allow us to fix the ordering $X\_1, X\_2, X\_3, X\_4$ for convenience.   We have one variable with zero parents, one with one, one with two, and one with three.  For zero and one parents, we have no hypergraphical choices.  For two parents, we may choose that the parents act independently {{1},{2}} or that they cooperate jointly {{1,2}}.  For three parents, we have 9 choices of hierarchical hyperparents.  One with all singles, three for each single pair, three for each pair of pairs, one for all three pairs, and finally one with the triple of parents. In conclusion, $432=24 \cdot 2 \cdot 9$.

---

> > > > > ### Author Response · Authors · 2025-08-07
> > > > >
> > > > > Altogether, we hope that the provided clarifications and method details will provide enough information for you to thoroughly evaluate the work.  We hope after this discussion you have an updated belief on the value of the submitted work, and we hope that the writing adjustments made will allow this value to be more easily appreciated by a wider audience. Overall, we feel the greatest concerns which might still remain are as follows:
> > > > >
> > > > > **W2**: Limited experimental evidence
> > > > >
> > > > > Although we hope to have convinced you above that this is not at all an issue with the quality of experiments (following standard approaches in the literature), we can understand why you take issue with the quantity of experiments.  We agree that including further experiments in the final version on slightly varied hyperparameter and DGP settings would strengthen the empirical claims made.
> > > > >
> > > > >  **W8**: No simple experiments investigating why the learning of higher interaction effects does not work
> > > > >
> > > > > For this, our original work took for granted that previous higher-order works have shown the statistical complexity of learning higher order interaction effects.  Following your experiment suggested above, we have been able to confirm this previously discovered reasoning also applies to our setting of causal discovery.  We agree that including a simple experiment like this which verifies the proposed reasoning helps to empirically strengthen the claims made.
> > > > >
> > > > > **W9**: No ablation study for the importance of key components of the proposed algorithm
> > > > >
> > > > > We agree understanding the HCAM algorithm better would be valuable, and lack thereof can be considered a weakness.  Unfortunately, we cannot imagine any simple ablations to run.  Comparing against a non-greedy strategy quickly faces computational limitations.  Since there are 34 HDAGs on 3 nodes, 2165 HDAGs on 4 nodes, and 1.7 million HDAGs on 5 nodes, checking the likelihood of all these possibilities is simply not a feasible approach.  On the contrary, injecting oracle knowledge into the HCAM algorithm also seems nontrivial, because we do not evaluate on a metric like accuracy, but directly evaluate on the algorithm's ability to recover the structure.  Directly providing this structure is counterintuitive for evaluation.  We hope that improvements in the writing make it clear that HCAM is not positioned to be the end of the algorithmic developments in this setting, but only the beginning.
> > > > >
> > > > >
> > > > > If you feel that the justification for these remaining concerns are sufficient to support the conclusions made by the paper, we hope that you will seriously reconsider the value of this work.  If this can be done with some leniency towards the confusions which are now eliminated in the updated writing, it would be even more appreciated.

---

> > > ### Comment · Reviewer_YZBu · 2025-08-07
> > > **Clarification on the number of runs**
> > >
> > > *Again, I have to politely ask for clarification on* **R9**:
> > > 1) If you used 3 datasets for each of the three runs, there were 9 in total. As a consequence, one would expect at least a single average value with decimals other than 0, 0.33 or 0.67, but there is none in Table 2,3 and 4. Why is that so?
> > > 2) Independent of this, please clarify whether a run refers to a single sampled graph with different sampled parameters or multiple different graphs. If only 3 sampled graphs were tested at all, you definitely need to report their in the appendix. Even for 9 graphs is good scientific practice to include them in the paper and not hidden in optional code files that were not submitted for review (their adjacency matrices do fit on a single page).
> > >
> > > *Question in the rebuttal:*
> > > > Remark R9 (number of runs) was not included in the mentioned response to the other reviewer. Only the checklist of the submission reveals that 3 independent runs were conducted. Further inspection of the decimal values in Table 2,3 and 4 suggest that only a single graph and parameter coefficient (=1 data set) for each run was considered. If this is correct: Which 3 graphs with 30 nodes were sampled? Please state the adjacency matrices in the appendix along with the sampled coefficients.
> > >
> > > *Answer in the rebuttal:*
> > > > For R9, we indeed use three different datasets for each run. Although adding all the 30x30 adjacency plots to the appendix seems a bit extreme, we can include the .json files with both the adjacencies and the multilinear coefficients to the code release, and because random seeds were set they should also be regeneratable.

---

> > > > ### Author Response · Authors · 2025-08-07
> > > >
> > > > Yes, in total there were 9 graphs generated (across three settings).  Each "run" was an independent run of the algorithm (this was of course repeated for each algorithm).  For each of the 3 settings, we took the mean and variance of those 3 evaluated metrics.  Interestingly, you are right that because each score is an integer value and we take 3 runs, this leads to each mean value ending in X.00, X.33, or X.67.  We had not noticed this before.

---

> ### Comment · Reviewer_YZBu · 2025-08-07
>
> Thank you for the final clarification that the reported mean values and standard deviations in Table 2, 3 and 4 are only taken over 3 datasets. The answer also clarifies that for the same graph, not multiple parameter values were sampled.
>
> As a consequence, the interpretation of the standard deviations using $N_\text{runs}=3$ and resulting claims have to be lowered by the authors accordingly. (*Only*) for the three same tested graphs in each setting, they allow for a direct comparison between the algorithms.
> Hence, the presented empirical evidence for each setting is very limited and certainly to weak to support general statements about the performance of the algorithms on the DGP.\
> This explains why I highly recommended to state the tested adjacency matrices and parameters.

---

> > ### Author Response · Authors · 2025-08-08
> >
> > We apologize, we do not understand the concern here.  Is there a different way to perform experiments which is preferred?
> >
> > Please let us know if you have any final questions on the other points raised since the discussion phase will end soon.

---

> > > ### Comment · Reviewer_YZBu · 2025-08-08
> > >
> > > The criticism of the results in Table 2,3,4 and their interpretation is their limited reliability given the very low number of samples.
> > > The DGP specifies a distribution over graphs and their parameter, but only 3 data sets were generated and used to estimate the mean and standard deviations. It is rather straight-forward to see that the empirical evidence is very weak due to the small data size.

---

> > > > ### Comment · Reviewer_YZBu · 2025-08-08
> > > >
> > > > I have no further questions, all questions of my initial review were sufficiently addressed by the authors in the rebuttal which is appreciated. However, the limitations, some weaknesses and concerns from the remarks remain.
> > > >
> > > > Here is the feedback on your explicit questions in the last comments:
> > > >
> > > > **Q2:**
> > > > > This confusion is especially concerning because we imagine that future work should be able to extend beyond the case of observed variables, as discussed in our original response to W1. Would a sentence warning against this presumption about LVs be sufficient (e.g. adding this note to line 105)?
> > > >
> > > > I suggested latent mediators (with special nodes and special edges as you correctly stated) as an alternative to the proposed hyperedges. While it is completely fair to say, that there are reasons to model hyperedges, I suggested briefly discussing the alternative for which [3] provides some relevant insights (although not the main focus of their work).\
> > > > A simple note that ANMs are known *not* to be closed under marginalization (of general latent variables) seems good.
> > > >
> > > >
> > > > **W7**:
> > > > > Is it preferred that we move line 249 to the beginning of the section? (...) We can add a sentence to Theorem 3 that it uses a similar proof technique to Theorem 2 (...).
> > > >
> > > > Yes, beginning with a reference to the proofs in the appendix is preferred and adding at least some intution/high level argumentation for Theorem 3 is appreciated.
> > > >
> > > > **R10**:\
> > > > I appreciate that the authors included the zero baseline from the very start. However, the authors should clarify that the zero baseline is not beaten only for the 3D data on the tested *sparse* ER4 model (if the results were statistically reliable). Other random graph model with a higher number expected edges per node were not tested.

---

> > > > > ### Author Response · Authors · 2025-08-09
> > > > >
> > > > > So, you are suggesting the results in Table 2,3,4 have an extremely strong dependence on the 3 sampled graphs (3 per row, 9 graphs in total)?  We believe looking at the scale of the variances compared to the mean should dissipate that concern.
> > > > >
> > > > > Personally, we believe that all of the remarks have been addressed.  We have already stated our opinion on L2 and L3, and we have also clarified which weaknesses we believe were the most important initial concerns (W2, W8, W9).  Nevertheless, we hope to have convinced you that these weaknesses have be reduced via new experiments and are overshadowed by productive aspects of the work.
> > > > >
> > > > > **Q2**
> > > > >
> > > > > The entire goal of [3] is to replace latent variables with a hypergraph representation.  To be specific, each hyperedge represents marginalization over a latent variable.  As stated before, an "MC-graph" like extension of our work would subsume both this work and [3].  Do you have a different interpretation of the main focus in [3]?
> > > > >
> > > > > **W7**
> > > > >
> > > > > "All full proofs..." is now at the end of line 199.  Line 240 now reads "The proof technique is the same as Theorem 2."
> > > > >
> > > > > **R10**
> > > > >
> > > > > We still believe asking for denser graphs disregards the status quo of the causal discovery literature.  Although we agree that one empirical setting (ER4 D30 N10000) limits generalizing the claims, we maintain that the setting is relevant and that our conclusions are valid.  Moreover, we feel the new experiments on different sample sizes confirm that our stated hypothesis is correct.

---

> > > > > > ### Comment · Reviewer_YZBu · 2025-08-09
> > > > > >
> > > > > > > So, you are suggesting the results in Table 2,3,4 have an extremely strong dependence on the 3 sampled graphs (3 per row, 9 graphs in total)?  We believe looking at the scale of the variances compared to the mean should dissipate that concern.
> > > > > >
> > > > > > I did not suggesting the alleged statement, we simply do not know that. For your claim and reasoning you have to prevent reliable experimental evidence.
> > > > > > 3 sampled graphs are insufficient to describe a whole distribution over graphs and compute reliable statistics. This is a simple matter of statistical analysis and not beliefs.
> > > > > > Mentioning 9 graphs is distracting from the discussion, since 3 different ones are tested in 3 different settings.

---

### Official Review · Reviewer_35ca · 2025-07-02

**Clarity:** 2
**Significance:** 3
**Originality:** 3
**Rating:** 3
**Confidence:** 4

**Summary:**

This paper introduces novel causal structure learning framework by incorporating higher order interactions. Instead of directed acyclic graphs, the more general directed acyclic hypergraphs are considered. Extending from the traditional pairwise results, the identifiability results of the higher order structures are provided. A greed algorithm reconstruct the higher order structures is proposed. Finally, the method is validated using several synthetic datasets.

**Questions:**

1. The current definition of HyperDAG only has one child. It is clear that a more general form will be to have multiple child. Would this be useful in terms of causal structure learning? How will this affect the technical results obtained in the paper?

2. The data in the experiments are $i.i.d.$. Would a similar greedy algorithm work for time-series data (let's assume stationarity)?

3. There are other higher order structures, such as directed simplicial complexes and cell complexes. Can these structures be exploited to infer causality from data similar to the hyper DAG?

**Ethical Concerns:**

["NO or VERY MINOR ethics concerns only"]

**Final Justification:**

The main issue with the paper is that many of its contributions are presented in an overly obscure manner, and both the related work and introduction sections are poorly written. Although the paper shows potential, I do not believe it is sufficient for acceptance at this stage.

**Limitations:**

Yes.

**Quality:**

3

**Strengths And Weaknesses:**

**Strength:** The structure and the flow of the paper is good. Slowly transitioning from the pairwise graphs to the higher order graphs to help the readers understand the setup. There are also many useful theoretical results discussed in the paper which can be useful for future work in this direction.

**Weakness 1:** It might be trivial, but a lot of the technical results lack references. For example, the definitions related to the pairwise models. Moreover, many technical terms such as SEM and body are only mentioned without definitions and clear references.

**Weakness 2:** The data used in examples or the equations used to generate them should be available and more explicit than what is currently in Section 5.

---

> ### Author Rebuttal · Authors · 2025-07-31
>
> ## Strengths
>
> > This paper introduces novel causal structure learning framework by incorporating higher order interactions.
>
> > There are also many useful theoretical results discussed in the paper which can be useful for future work in this direction.
>
> ## Writing Weaknesses
>
> > W1: It might be trivial, but a lot of the technical results lack references. For example, the definitions related to the pairwise models.
>
> Thank you for bringing this to our attention, please see the “minor points” section of our response to Reviewer “nSx3” where we make point-by-point improvements to the writing based on their feedback about this writing issue.
>
> > W2: The data used in examples or the equations used to generate them should be available and more explicit than what is currently in Section 5.
>
> We have added more of these details to the appendix.  The data is generated as follows.  We first sample the ER graph and then sample a random ordering of nodes (it does not matter which order we do this in).  In the 1D setting our HDAG is already complete.  In the 2D and 3D setting, we choose hyperedges by taking a random ordering of parents and selecting pairs or triples in a cyclic fashion.  This is the simplest HDAG which obeys the same DAG structure. Now we sample the parameters.  For the beta coefficients, we sample from [0.5, 2.0] using a log uniform distribution.  We sample gaussian standard deviations from the same distribution. We then rescale the coefficients by dividing by the square root of the expected Gaussian moments, 3.0 and 15.0, respectively, and the divide by the total number of parents as well.  For the 1D case we follow the literature standard of sampling from a Gaussian process prior (randomly initialized sklearn.gaussian_process.GaussianProcessRegressor). Code will be released.
>
> ## Question – Extendable to Time Series?
>
> > Q2: The data in the experiments are i.i.d. Would a similar greedy algorithm work for time-series data (let's assume stationarity)?
>
> There are some existing works showing the identifiability of causal structure in time series settings [E, F].  Given that our theoretical results show that HDAGs are always identifiable when the DAGs are identifiable, we expect the same would hold for any TS generalization.  If your question is about the greedy algorithm, already for our 3D experiments the algorithm is struggling, which would only become more challenging in TS.  This might helps explains why TS researchers already make low-dimensional assumptions (e.g. channel independence) in practice.
>
> ## Question – Other Algebraic Structures?
>
> > Q3: There are other higher order structures, such as directed simplicial complexes and cell complexes. Can these structures be exploited to infer causality from data similar to the hyper DAG?
>
> Yes, if by complexes you are referring specifically to abstract simplicial complexes, then the structure defined by an ASC is exactly the same as the structure defined by the undirected hypergraph.  In some sense, it would be more accurate to introduce our framework using ASCs because we already assume the hierarchy property or “closure under subsets” property of the hypergraph edges.  However, we felt it would be much more natural to extend the existing graphical language which has been built up over decades.
>
>
> > Q1: The current definition of HyperDAG only has one child. It is clear that a more general form will be to have multiple child. Would this be useful in terms of causal structure learning? How will this affect the technical results obtained in the paper?
>
> No, lifting the restriction to use hyperedges which have a single child (single arrowhead) does not have any use in this framework, as mentioned next to its definition.  Because of the ordering on the variables coming from the DAG framework, we will only generate one variable at a time.  Accordingly, if one tries to have a hyperedge with k children, it will be equivalent to splitting the hyperedge into k hyperedges with one child, only making the results of the paper less clear.
>
>
> One could imagine multiple arrowheads being useful in an extension to latent variables such as in mixed graphs where bidirected edges are often used to represent confounding.  However, the complexity of such an extension skyrockets.  It took a decade of research in the 90s to go from DAGs to MAGs and exactly classifying the equivalence of different mixed graphs remains an open problem and an active area of research.  For why latents are hard even in our CAM setting, see [G].
>
>
> All that is to say, (a) for our paper’s setting without latents, it is not useful, and (b) for a setting with latents, it could be much more complex than “HDAGs with multiple children”.
>
> [E] “Search for Additive Nonlinear Time Series Causal Models”. Tianjiao Chu and Clark Glymour. 2008.
>
> [F] “Causal Inference on Time Series using Restricted Structural Equation Models” Jonas Peters et al. 2013.
>
> [G] “Causal additive models with unobserved variables” Takashi Nicholas Maeda, Shohei Shimizu. UAI 2021.

---

> > ### Comment · Reviewer_35ca · 2025-08-01
> >
> > Thank you to the authors for the rebuttal.
> > 1.  I agree that the hypergraph is a more natural extension. Still, I believe the link to simplicial complexes should be briefly discussed as a remark, as this is a popular representation in the higher-order literature and would help readers quickly understand the setting of this paper.
> > 2.  The time series extension and the ordering issue in the greedy algorithm both point to the same scalability challenge of incorporating higher-order components. Some analysis of time complexity or practical implementation issues could be a valuable addition to the paper.
> > 3.  I appreciate the textual description of the data generation process. Including some equations would make it more straightforward, especially in conjunction with the eventual code release.
> >
> > Overall, I believe this is a very pioneering piece of work. Currently lots of the useful stuff is hidden or implicit, as noted by other reviewers. I understand this may be due to the page limitations of ML conferences. By addressing these points and having a clearer presentation, I believe this can be a very strong paper.

---

> > > ### Author Response · Authors · 2025-08-01
> > >
> > > 1. We are glad you agree that hypergraphs are the most natural algebraic structure for this
> > >
> > > 2. We believe that the fact that higher-order components are difficult to incorporate, exemplified by the (imaginary) difficulties for time series extensions and by the simplistic choice of using a greedy algorithm, is a fact which is fundamentally true.  Taking for granted that higher-order interactions do exist in real-world data, we feel it is important to first demonstrate this necessity before solving all of the many algorithmic challenges.  Importantly, the existing algorithms which simply choose to ignore these aspects are not safe either, as demonstrated by their poor performance.  We apologize that this feeling was not able to come across in the writing.  Hopefully some further discussion on the practical implementation issues would enhance the reader's understanding of this difficulty, allowing them to better appreciate this higher-order setting.
> > >
> > > 3. Great
> > >
> > > Thank you for your interesting review

---

### Official Review · Reviewer_iobK · 2025-07-02

**Clarity:** 3
**Significance:** 2
**Originality:** 2
**Rating:** 3
**Confidence:** 4

**Summary:**

This work studies the problem of causal discovery in additive models with higher-order interactions. Such interactions are depicted via directed acyclic hypergraphs.  Lastly, it develops a greedy algorithm to learn such hypergraphs and evaluates it in synthetic experiments.

**Questions:**

Please see the comments.

**Ethical Concerns:**

["NO or VERY MINOR ethics concerns only"]

**Final Justification:**

The primary concern about the method’s applicability and scalability remains, and therefore my recommendation is still to reject.

**Limitations:**

Please see the comments.

**Paper Formatting Concerns:**

No major formatting issues.

**Quality:**

2

**Strengths And Weaknesses:**

Strength:
Using Hypergraphs for representing higher-order interactions is useful and can be used in various fields when such type of interactions are as important as knowing cause-effect relationships.

Extended identifiability results for hypergraphs are interesting and might be valuable for further researches.

Weakness:
The main concern is about the usefulness of the hypergraph and higher-order interactions. It is still not so clear in what type of applications knowing (10) is better than (9).
For instance, let x_1 = x_2 + x_3 + x_4 + e_1. This can be written in various forms with different hypergraph representations but all following the form in (10), e.g.,
x_1 = (x_2 + x_3)/2 + (x_4)/2 + (x_2 + x_3 + x_4)/2 + e_1
x_1 = (x_2)/3 + ((x_2)/3 + x_4) + ((x_2)/3 + x_3) + e_1
This not only affect the identifiability of the problem but also raises the application of the model, meaning that in what application distinguishing these different formulations is relevant.

Eq. (4) is not clear. If \xi_S(x_s) is the log probability of p(x_s), then (4) is not correct. According to the Hammersley-clifford theorem, the joint probability can be represented using the cliques but the terms are not necessary the probabilities.


Regrading the proposed greedy method, does its complexity grow with the order of interactions? This affects its scalability. Moreover, the order of interactions should be a hyper parameters, how is that selected?

---

> ### Author Rebuttal · Authors · 2025-07-31
>
> ## Strengths
>
> > Using Hypergraphs for representing higher-order interactions is useful and can be used in various fields when such type of interactions are as important as knowing cause-effect relationships.
>
> > Extended identifiability results for hypergraphs are interesting and might be valuable for further researches.
>
> ## Weaknesses
>
> > The main concern is about the usefulness of the hypergraph and higher-order interactions. It is still not so clear in what type of applications knowing (10) is better than (9).
>
>
> Knowing (10) is true is always strictly better than knowing (9) is true, containing **strictly** more information about the structure.  It is in this sense that we say the HDAG structure strictly extends the DAG structure without making any parametric SEM assumptions, like Eqn (1).  Whenever the DAG is identifiable, we show that **without** any additional assumptions, the HDAG is also identifiable.  Therefore, any time you can know (9), you can also know (10).  This is the importance of our identifiability results both in classical DAGs and in ANMs.
>
> All of these rewritings of $x_1 = x_2 + x_3 + x_4 + e_1$ have the exact same HDAG structure according to equation (10).  Namely $x_1 = f(x_2) + f(x_3) + f(x_4) + e_1$.  A slightly more interesting example could be $x_1 = log( x_2 \cdot x_3 \cdot x_4) + e_1$ which **does not** have the HDAG structure $x_1 = f(x_2,x_3,x_4) + e_1$, but instead we can see that $x_1 = log(x_2) + log(x_3) + log(x_4) + e_1$, so we have the same $x_1 = f(x_2) + f(x_3) + f(x_4) + e_1$ structure as before.
>
> > Eq. (4) is not clear. If \xi_S(x_s) is the log probability of p(x_s), then (4) is not correct. According to the Hammersley-clifford theorem, the joint probability can be represented using the cliques but the terms are not necessary the probabilities.
>
> You are correct that Equation (4) is essentially the Hammersley–Clifford theorem and that in general the $\xi_S(x_S)$ is not equal to $\log(p(x_S))$.  The same is true of Equation (5) below it.  We apologize if this is confusing due to us defining $\xi_X$ and introducing the clique representation using $\xi_S$ in the same line.
>
> > Regarding the proposed greedy method, does its complexity grow with the order of interactions? This affects its scalability. Moreover, the order of interactions should be a hyper parameters, how is that selected?
>
> You are correct, the complexity grows with the order of interactions.  In particular, we take the maximal possible degree of an interaction as the hyperparameter (set equal to three throughout our experiments).  This means the greedy, layerwise approach taken by SIAN is able to stop early if the real data has no interactions, but cannot add any interactions of degree 4 or higher.  Setting the maximal degree to something larger like 5 would slow down phase 1 of the algorithm and also create a larger candidate set for phase 2.
>
> These questions of how to learn higher-order interactions while balancing scalability are not addressed anywhere in current literature and we believe this contributes to most algorithms’ failure in Tables 2, 3, and 4.  In the real world, the *true degree* will depend on the actual causal mechanisms behind the data generation.  We believe this tradeoff is an important part of causal learning which is not shown in current literature, even though higher-order interactions are already known to exist from physics [A, B, C].

---

> > ### Comment · Reviewer_iobK · 2025-08-04
> >
> > Thank you to the authors for the rebuttal. However, my concerns regarding the applicability and scalability of the method remain. While the work has merit, I believe it will be significantly strengthened by addressing the reviewers' concerns. Therefore, I am maintaining my original score.

---

> > > ### Author Response · Authors · 2025-08-04
> > >
> > > We are sorry that we were not able to alleviate your concerns regarding applicability and scalability.  We agree the work will be strengthened by following other reviewers' suggestions and by also addressing your concern "does its complexity grow with the order of interactions? This affects its scalability." more explicitly in writing.

---

### Author Response · Authors · 2025-08-07
**Summary**

We thank all reviewers for their feedback on our work.  Based on the advice of reviewer “YZBu”, we will summarize the discussion between all reviewers and in particular summarize the changes made to the draft.  It is hoped that this summary will help enable an accurate assessment of the work and the claims being made, especially after the improvements to the clarity of presentation.

We begin by reemphasizing the strengths.  All four reviewers identify the new settings and the new theoretical results as key strengths of the work.  Given the ratings, however, we are forced to assume that we disagree with the reviewers on the novelty and the significance of these contributions.  To be straightforward, the authors believe that this is the first extension of DAG identifiability in the fully observed case since Pearl and Verma’s original 1990 result characterizing the Markov equivalence classes. Although we do not claim there are zero empirical weaknesses, we hope the significance of these results can be considered as a larger factor in the final decision.

Before discussing the changes to writing and presentation which we believe resolves many of the misunderstandings from the discussion phase, we first quickly reiterate the largest remaining concerns which were not completely resolved during the discussion and our response to them.


## Remaining Concerns

- Scalability concerns on the proposed algorithm

  - We want to emphasize that we feel the setting we introduce is fundamentally hard, rather than a property of the algorithm.  Now, unlike previous works, we know which part is hard (the higher-order interactions).  Accordingly, while we agree there are limitations with the proposed algorithm, it is currently the best algorithm for this setting.  We hope better writing can clarify this.

- No ablation study for the key components of the proposed algorithm

  - Although we agree in principle that it would be useful to understand the HCAM algorithm in greater detail through ablation studies, we do not believe it is feasible to do so.  Because we are introducing a new setting, there are no alternative algorithms to ablate with. Alternatively, providing oracle information about the structure seems counterintuitive to our evaluation on recovering the structure.  Lastly, comparison against a non-greedy version is not computationally feasible.  The number of possible hypergraphs grows doubly exponentially and with D=5 nodes, there are already more than one million hypergraphs.

- Insufficient experimental evidence
  - This critique took multiple forms: only simple data generating processes with sparse graphs, no use of real-world experiments, and insufficient experiments explaining why the algorithms failed.  We hope that it is more clear than before why at least some synthetic data is absolutely necessary to evaluate structure learning algorithms.  The inability to use real-world data is compounded by our findings which show that existing algorithms struggle to uncover higher-order interactions, even for simple data and even for our algorithm specifically designed to do so.  Accordingly, for the negative conclusion of our work, we still believe that the simplicity of the data generation is only an advantage for our findings.  With all of that as context, we do have new experiments coming from our discussion with reviewer “YZBu” which clearly demonstrate that increasing the number of samples allows the HCAM algorithm to learn the higher-order mechanisms.  We hope this helps provide additional experimental evidence proving that our claimed reasoning is correct.

---

> ### Author Response · Authors · 2025-08-07
>
> We hope that most of these concerns can be alleviated through better writing, and also through better positioning of how our claims fit into the current literature on causal structure learning.  To enable a better comparison, we include a list of the major changes to improve the writing and it is hoped that this will allow for the possibility to reconsider the value of the proposed work.
>
>
> ## Major Writing Changes
> - Added further motivation on why higher-order is known to be important for real-world systems.
>   - The paragraph at line 40 now reads: “In this work, we revisit the additive structural assumption of CAM to also incorporate higher-order interactions, extending the causal model to explicitly consider higher-order causal mechanisms. Higher-order mechanisms are known to exist in a variety of real-world processes [Battiston et al., 2020], and are critical for modeling a number of different scientific phenomena including dynamical systems [Majhi et al., 2022], bioscience [Taylor and Ehrenreich, 2015, Gaudelet et al., 2018], neurobiology [Amari et al., 2003, Petri et al., 2014], and social networks [Freeman and White, 1993, de Arruda et al., 2020]. Nevertheless, previous approaches to causality fail to adequately represent the higher-order mechanisms which could be at play in real-world data. Most algorithms take an all-or-nothing approach, either (a) directly following CAM-like assumptions of no interactions or (b) modeling all possible interactions between the parents of a child node.”
> - We add to the end of contribution 2 at line 56, “Moreover, experiments on simple synthetic data demonstrate that most algorithms struggle to capture higher-order interactions, calling into question the practical learnability of real-world systems with higher-order interactions.”
> - We move the paragraph at line 79 to just above Section 2.1
> - Added references to existing definitions (defn 1, 3, 5) and existing experiment settings (line 299)
> - Definition, clarity
>   - Line 118 now reads: “It is straightforward to see that this again strictly generalizes the structures which are representable by the typical DAG.  Importantly, this includes structures which are not easily represented through existing latent variable approaches.  It can also be seen from equation 8 that we may restrict our attention to hypergraphs which are *hierarchical*, meaning $S \\in \\text{HypPa}\_{\\cal{H}}(j)$ and $T \\subseteq S$  implies that $T \\in \\text{HypPa}\_{\\cal{H}}(j)$ (much like a simplicial complex).”
> - Results, additional sentences
>   - “We bold those algorithms which beat the zero baseline.  In the case of no statistically significant outperformance, we bold algorithms which perform no worse than the baseline.”
>   - “In several cases, BOSS on 2D, and BOSS, GES, HCAM on 3D, we find that the performance of the baseline is matched by directly predicting the zero baseline (partially or completely).”
>   - “In Figure 3, we see how the number of provided samples directly affects the ability to learn higher-order mechanism structure.  Even for this small hypergraph, learning with 100K samples is not sufficient for perfect recovery 100% of the time.”
>
>
> ## Major Presentation Changes
>
> - Added a definitions section to the appendix
> - Improved Figure 1 with depicted SEMs (and updated caption)
> - Improved Figure 2 visually (and updated caption)
> - Added all data generation details to appendix
> - Added an experiment explicitly verifying that dataset size and scaling *is* the key issue
> - Added algorithm pseudocode to the appendix
>
> [H] “Higher-order molecular organization as a source of biological function” Gaudelet et al. 2018.
> [I] “Synchronous firing and higher-order interactions in neuron pool” Amarai et al. 2003.
> [J] “Homological scaffolds of brain functional networks” Petri et al. 2014.
> [K] “Using Galois Lattices to Represent Network Data” Freeman and White. 1993.
> [L] “Social contagion models on hypergraphs” de Arruda et al. 2020.

---

### Note · Authors · 2025-08-14

We would like to thank everyone for their reviews one more time.  Although we do not have much new information, we hope that many of the initial points of confusion have been resolved by pointing to references in the existing literature.  We also hope that other reviewers will now consider raising their score after >90% of the concerns from reviewer "YZBu" have been addressed.  In our global comment which provided a summary of changes, we detailed the writing changes which will improve clarity of presentation and we explained why the remaining concerns are minimal.  One last time, it is not possible to evaluate causal discovery metrics on real-world data and the critique on "3 graphs" is not a valid statistical concern.  Reporting the average performance across three or five runs is a completely standard procedure taken by many Neurips papers.  If you would like to criticize our algorithm, please remember that nearly every published algorithm is doing strictly worse (except CAM and SCORE, which are also greedy approaches.)  Finally, given they are the key strengths of the paper, we hope that the *originality* of introducing a new causal setting and proving the identifiability as well as the *significance* of the unique conclusions offered by this paper which have the potential to impact future works, are taken into greater consideration during the final reviews and decision.

---

### Decision · Program_Chairs · 2025-09-17

**Decision:**

Reject

**Comment:**

This paper presents an extension of causal additive models to handle higher-order interactions, introducing a hyper-DAG formulation, identifiability results, and an extended CAM algorithm. The topic is interesting and potentially valuable to the causal discovery community. However, all reviewers gave negative evaluations. They raised significant concerns about clarity of presentation, insufficient discussion of related work, and weak empirical validation. Questions about the applicability and scalability of the method, as well as the lack of real-world experiments, further limit its impact. While the rebuttal addressed some points, substantial improvements remain necessary. Given the highly selective nature of this venue, I recommend rejection in its current form.